# CITYANCHOR: CITY-SCALE 3D VISUAL GROUNDING WITH MULTI-MODALITY LLMS

**Jinpeng Li**[1][*] **Haiping Wang**[1][,][*] **Jiabin Chen**[1] **Yuan Liu**[2][†] **Zhiyang Dou**[3] **Yuexin Ma**[4]
**Sibei Yang**[4] **Yuan Li**[5] **Wenping Wang**[6] **Zhen Dong**[1][,] **Bisheng Yang**[1][,][†]
[1]LISMARS, Wuhan University  [2]Hong Kong University of Science and Technology
[3]University of Pennsylvania [4]ShanghaiTech University  [5]Sun Yat-Sen University
[6]Texas A&M University

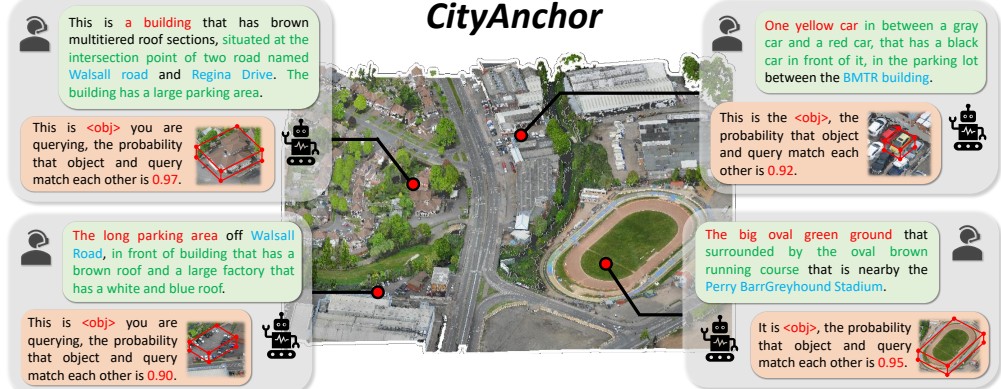

Figure 1: We present CityAnchor, a multi-modality LLM, that can accurately localize a target in a city-scale point cloud from some text descriptions of the target. CityAnchor achieves this by extracting features from the point cloud to grasp the intricate attributes and spatial relationships of urban objects. Then, CityAnchor comprehends the text descriptions and searches in the urban-scale point cloud for the objects corresponding to the input text descriptions.

## ABSTRACT

In this paper, we present a 3D visual grounding method called CityAnchor for localizing an urban object in a city-scale point cloud. Recent developments in multiview reconstruction enable us to reconstruct city-scale point clouds but how to conduct visual grounding on such a large-scale urban point cloud remains an open problem. Previous 3D visual grounding system mainly concentrates on localizing an object in an image or a small-scale point cloud, which is not accurate and efficient enough to scale up to a city-scale point cloud. We address this problem with a multi-modality LLM which consists of two stages, a coarse localization and a fine-grained matching. Given the text descriptions, the coarse localization stage locates possible regions on a projected 2D map of the point cloud while the fine-grained matching stage accurately determines the most matched object in these possible regions. We conduct experiments on the CityRefer dataset and a new synthetic dataset annotated by us, both of which demonstrate our method can produce accurate 3D visual grounding on a city-scale 3D point cloud. The source code is available at https://github.com/WHU-USI3DV/CityAnchor.

## 1 INTRODUCTION

3D visual grounding is a critical task in computer vision with transformative applications in robotics, AR/VR (Anderson et al., 2018; Lee et al., 2021), and autonomous driving (Najibi et al., 2023). Taking this to the next level by scaling 3D visualization to city-scale point clouds opens up thrilling new possibilities. Imagining the power to analyze entire cities in detail, we could revolutionize

---

[*]The first two authors contribute equally.
[†]Corresponding Authors.

urban planning and infrastructure development. This leap in scale can propel us into a new era of city-wide analysis, enhancing decision-making processes and fostering innovative solutions for urban challenges. Accurate city-scale 3D visual grounding is set to inspire the next generation of map products and drive advancements in GeoScience, offering an unprecedented view of urban environments.

Scaling 3D visual grounding to a city scale is a challenging task. Existing work CityRefer (Miyanishi et al., 2023) has tried to tackle this task by creating a large-scale point cloud visual grounding dataset and training a neural network to embed both text descriptions and point clouds within the same feature space. Then, in inference time, given the text description and 10 possible candidate objects, CityRefer exhaustively compares the provided candidate objects to find the best match, which demonstrates impressive performance in this city-scale visual grounding task. However, training a multi-modality network from scratch to extract features for both text descriptions and point clouds shows the limited capacity for complex input texts. Meanwhile, CityRefer requires 10 candidates as inputs and exhaustively matching all the objects in the city is too time-consuming. Thus, how to improve the multi-modality feature extraction in the large-scale visual grounding and how to efficiently localize the object in a large-scale point cloud remain two open problems.

A promising direction to improve visual grounding is to design a multi-modality Large Language Model (LLM) (Touvron et al., 2023; Lai et al., 2023; Hong et al., 2023b) to process both the text prompts and 3D point clouds. LLMs show a strong ability to understand the text descriptions and by aligning the features of point clouds with the feature space of LLMs, we are able to conduct visual grounding with the help of LLMs. There are already some pioneer works (Yang et al., 2023b; Liu et al., 2024a; Yang et al., 2023a;a; Zhu et al., 2023; Jia et al., 2024) following this path to incorporate LLMs with point clouds for the small-scale 3D visual grounding task. Most of these works concentrate on visual grounding in small-scale indoor point clouds containing less than 10 objects so they can exhaustively match each object with the text description by their multi-modality LLMs. However, such an exhaustive matching is too costly for a city-scale visual grounding with hundreds or thousands of objects.

In this paper, we propose CityAnchor for the city-scale 3D visual grounding task, as shown in Fig. 1. CityAnchor addresses this challenging large-scale visual grounding task with two designs. First, we finetune construct a multi-modality LLM based on a pretrained LLM (Liu et al., 2024b) to simultaneously process the 2D maps, 3D point clouds, and text descriptions. By aligning the features of 2D maps and 3D point clouds with the well-established language feature space, CityAnchor is able to accurately find the target objects and generalize well to unseen complex text descriptions. Second, we design a coarse-to-fine searching strategy for efficient visual grounding among hundreds or thousands of objects in a city-scale scene. The coarse searching strategy of CityAnchor efficiently identifies a set of possible regions for the target objects by predicting a heatmap on the projected 2D maps. Then, the objects in these possible regions are extracted and compared with text descriptions to predict similarity scores. This coarse-to-fine searching strategy greatly reduces the number of fine-grained comparisons by LLMs and enables our method to scale up to urban point cloud.

To evaluate the performance of CityAnchor, we conduct experiments on the CityRefer dataset and a new synthetic self-annotated dataset. We directly evaluate the visual grounding ability of CityAnchor in the city scale rather than selecting from 10 candidate objects. Results demonstrate that CityAnchor achieves state-of-the-art 3D visual grounding performance on the public CityRefer (Miyanishi et al., 2023) dataset and a self-annotated dataset, surpassing the baselines by $30\% \sim 43\%$ in grounding accuracy (Acc) on average. Moreover, CityAnchor requires only $\sim 32$ seconds to search for an object in a city-scale point cloud with more than $\sim 300$ objects, which is $1.3\times$ faster than the baseline CityRefer. The improved performances and efficiency in such a city-scale visual grounding can greatly benefit downstream geoscience applications to analyze large-scale scenes.

## 2 RELATED WORK

**Grounding in indoor scenes.** ScanRefer (Chen et al., 2020) and Referit3D (Achlioptas et al., 2020) first propose indoor datasets for 3D visual grounding that contain free-form object-annotation pairs on ScanNet (Dai et al., 2017) dataset. On this basis, numerous point cloud-based visual grounding efforts have subsequently sprung up. Early works (Huang et al., 2021; Gu et al., 2023) resort to manually designed scene graph construction methods to delineate spatial relationships among object

proposals for locating. Recent methods (He et al., 2021; Roh et al., 2022; Zhao et al., 2021; Huang et al., 2022; Yang et al., 2024; 2021; Chen et al., 2023b; Guo et al., 2023) have pivoted towards the development of large transformer networks (Vaswani et al., 2017) for visual encoding, text-visual feature alignment, and visual grounding. With the ascension of Large Language Models (LLMs), there has been a burgeoning interest (Chen et al., 2023a; Yang et al., 2023a; Hong et al., 2023b;a; Fu et al., 2024; Chen et al., 2024; Zhang et al., 2024; Li et al., 2024; Tang et al., 2024) in exploring the capabilities of LLMs to interpret the complex 3D indoor scenes for 3D visual tasks such as grounding and planning. However, the above methods designed for indoor scenes can not naively scale up to city-scale grounding task for their limited grounding capability or memory constraints.

**Grounding in city-scale scenes.** Research towards 3D visual grounding for open outdoor scene remains nascent. Earlier works including TouchDown (Chen et al., 2019) and KITTI360Pose (Kolmet et al., 2022) aim at grounding in point clouds collected by a vehicle-based system, which is constrained to roadside environments. CityRefer (Miyanishi et al., 2023) annotated the large-scale SensatUrban (Hu et al., 2021b) dataset to train a simple baseline for city-scale 3D visual grounding. In CityAnchor, we leverage the strong knowledge prior from pre-trained LLMs to achieve much better performances.

**Multi-Modality Large Language Models.** Large Language Models (LLMs) like GPT-4 (Achiam et al., 2023) and LLaMA (Touvron et al., 2023), extensively trained on vast textual datasets through self-supervised learning paradigms, exhibit versatile capabilities in addressing diverse language-related tasks while demonstrating robust generalization capacities. Inspired by the exceptional reasoning capabilities of LLMs, researchers are actively investigating methodologies to extend these competencies into understanding and generating other modalities (Wang et al., 2024; Lai et al., 2023; Hong et al., 2023b; Yang et al., 2023a; Chen et al., 2023a; Ma et al., 2024; Yin et al., 2024; Wu et al., 2023a), including visual and three-dimensional spatial contexts, called multi-modality LLMs. CityAnchor is built upon the well-known MM-LLM LISA (Lai et al., 2023) and 2D vision LLM LLaVA (Liu et al., 2024b).

## 3 METHOD

### 3.1 OVERVIEW

CityAnchor aims to locate a 3D target within a city-scale colored point cloud $S$ based on a given text description $t$. CityAnchor is a multi-modal large language model (LLM) that operates in two stages: a coarse localization stage and a fine-grained matching stage. In the data preprocessing stage, we first segment $S$ into objects using a pretrained 3D segmentation model (Vu et al., 2022) and assign the street or building names, so-called landmarks, obtained from OpenStreetMap to each object following CityRefer (Miyanishi et al., 2023). Then, in the first stage, CityAnchor employs a Coarse Localization Module (CLM) to regress the possible regions of the target on a 2D map projected from the city point cloud, effectively filtering out irrelevant objects. In the second stage, CityAnchor uses a Fine-grained Matching Module (FMM) to perform fine-grained comparisons between each candidate object in the above regions and the text description, ultimately selecting the most similar object as the grounding result.

### 3.2 COARSE LOCALIZATION STAGE

The coarse localization stage of CityAnchor is shown in Fig. 2. Given a city-scale colored point cloud $S$, we first project the point cloud onto the XoY plane to obtain a 2D map (Liu et al., 2024a), denoted as $I$. Subsequently, both $I$ and $t$ are fed into a so-called Coarse Localization Module (CLM), which then predicts a heatmap indicating the possible locations of the target object.

The architecture of CLM follows LISA (Lai et al., 2023), including a 2D vision LLM initialized from LLaVA (Liu et al., 2024b), and an image segmentation model initialized from SAM (Kirillov et al., 2023). We extend the original vocabulary of LLaVA with a new token, called $< \mathrm{RoI} >$, the abbreviation for "Region of Interest".

Specifically, we first feed the projected 2D map $I$ and the text description $t$ into CLM and fine-tune the CLM to produce output texts $T_c$ that include the $< \mathrm{RoI} >$ token. Next, we extract the last-layer feature in the LLaVA of the CLM model corresponding to the $< \mathrm{RoI} >$ token. This feature, along

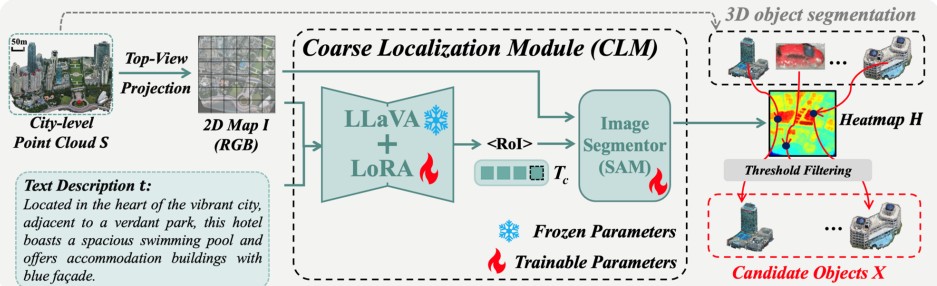

Figure 2: Coarse localization stage of CityAnchor, where CityAnchor accepts a text description and a 2D map projected from the city-scale point cloud and predicts a heatmap of the correlation score with the input text descriptions. The heatmap helps us to filter out a set of candidate objects.

with $I$, is input into the image segmentation model to generate a heatmap $H$ that contains the correlation score of every pixel. Then, for each target in $S$, we compute its correlation score using the score of its center point in $H$. If this value is greater than the threshold $\theta$, we consider it a candidate object. All candidate objects together form a set of candidates $X$.

**Discussion.** This CLM takes advantage of the strong feature extraction ability of LLaVA to understand both the input text descriptions and the image, which improves the generalization ability to unseen complex input texts than a vanilla feature extraction network. The regressed heatmap filters out irrelevant objects and enables concentrating on just relevant objects.

### 3.3 FINE-GRAINED MATCHING STAGE

Given a candidate object $x \in X$ and the text description $t$, we train a multi-modality LLM as our Fine-grained Matching Module (FMM) to predict the similarity between them. Then, the object with the largest similarity is selected as the final grounding result for the text description. Our FMM is also adapted and fine-tuned from LLaVA (Liu et al., 2024b). The original LLaVA is designed to handle text and 2D image inputs rather than the 3D visual grounding of point clouds. We thus modify LLaVA to enable it to process 2D maps, 3D point clouds, and text descriptions as inputs and we design a novel output module on the output of LLaVA to predict the similarity score between the text descriptions and the 3D object point cloud.

#### 3.3.1 OBJECT ENCODING

To enable FMM to accept the candidate object $x$ and the text description $t$ for comparison, we need to encode the feature of $x$ and align it with the well-established feature space of LLaVA. Specifically, we tokenize and embed the input text description $t$ to a set of feature vectors $E_{txt} \in \mathbb{R}^{l \times d}$ as the original LLaVA, where $l$ is a pre-defined sentence length and $d = 1024$ is the feature dimension.

For one candidate object $x \in X$ represented by a point cloud, we extract three kinds of features for this object as the input to the LLaVA.

- *Attribute feature.* Attribute features are the encoded features about the attributes of the object like colors and shapes. Since LLaVA enables CLIP features as input, we also want our attribute features to be aligned with the CLIP feature space. To achieve this, we adopted two methods to encode the attribute features of input object $x$. First, we adopt Uni3D (Liu et al., 2024a) to directly extract CLIP features from 3D point clouds. We sample 4096 points from $o$ and feed them into Uni3D to obtain the 3D feature $E_x^g \in \mathbb{R}^{1 \times d}$. Second, we project the point cloud of $x$ to the XoY plane to get an image and feed it to the image encoder of CLIP (Radford et al., 2021) model to obtain a projected 2D feature $E_x^s \in \mathbb{R}^{c \times d}$, $c$ denotes feature length of 2D CLIP feature.
- *Landmark feature.* An object in a city typically has its own name, such as "Clare College", which we call landmark. Landmarks are frequently mentioned and critical in the city-scale visual grounding. For example, we will describe a building "next to Clare College". Therefore, if the object $x$ has a landmark name, we embed the name into $E_x^l \in \mathbb{R}^{1 \times d}$ using a BiGRU (Chung et al., 2014) model and set $E_l$ to zeros otherwise.
- *Spatial context feature.* In a visual grounding task, text descriptions often include relationships between the target object and its neighboring objects. To provide such spatial context information,

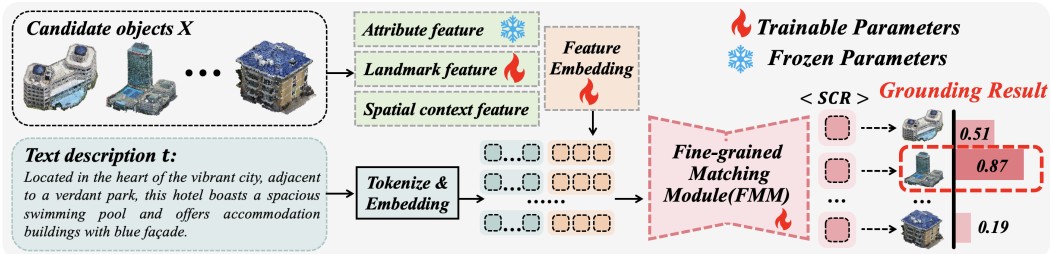

Figure 3: Fine-grained matching stage of CityAnchor. CityAnchor scores the similarity between the text description and each candidate object with the LLM-driven Fine-grained Matching Module. The object with the highest similarity score is chosen as the grounding result.

we extract $K$ nearest neighboring objects of $x$ denoted as $Y_x = \{y_i, i = 1, .., K\}$ and collect their attribute features and landmark features as $\{E_y^g | y \in Y_x, E_y^g \in \mathbb{R}^{K \times d}\}$ and $\{E_y^l | y \in Y_x, E_y^l \in \mathbb{R}^{K \times d}\}$ for $x$.

Then, we concatenate the above feature into a feature vector $\hat{E}_x = E_x^s \| E_x^g \| E_x^l \| E_y^g \| E_y^l \in \mathbb{R}^{(2+c+2K) \times d}$, where $\cdot \| \cdot$ means the concatenating on the feature dimension. Then, we further concatenate it with the text embedding $E_{txt} \in \mathbb{R}^{l \times d}$ to get a feature vector of length $(2 + c + 2K + l) \times d$, which serves as the input to the transformer encoder of FMM.

### 3.3.2 SIMILARITY SCORE PREDICTION

To enable FMM to predict the similarity between a text description $t$ and an object $x$, we expand the vocabulary of LLaVA with a new placeholder token, denoted as $< \text{SCR} >$. Subsequently, we train FMM to generate output texts $T_f$ containing the $< \text{SCR} >$ token. We then extract the last layer feature in FMM corresponding to the $< \text{SCR} >$ in the response. This feature is fed into an MLP to regress a score $s$, quantifying the similarity between the text description and the input object.

### 3.4 TRAINING LOSSES

The training of CityAnchor involves two stages. For CLM, we adopt an auto-regressive cross-entropy (ACE) loss (Liu et al., 2024b) to ensure that LLaVA generates reasonable text responses $T_c$ containing $< \text{RoI} >$. We further introduce a segmentation loss (Lai et al., 2023) to supervise the heatmap $H$ output by the segmentation model in CLM. The loss for CLM training is formulated by

$$L_{CLM} = \lambda_{txt} L_{txt}(T_c, \hat{T}_c) + \lambda_{seg} L_{seg}(H, \hat{H}), \tag{1}$$

where $\hat{T}_c$ is the annotated ground truth text answer containing $< \text{RoI} >$ token, $\hat{H}$ denotes the ground truth binary heatmap for candidate object selection in CLM with the ground truth region of the grounding target set to 1 and other pixels set to 0.

For FMM, we also adopt the ACE loss to ensure it generates text answers $T_f$ with the $< \text{SCR} >$ token. Then, we use the binary cross-entropy loss to supervise the regressed score $s$ of FMM to be the same as the ground truth score $\hat{s}$, where $\hat{s}$ is set to 1 if the input object corresponds to the text description and 0 otherwise. The loss for FMM training is formulated by

$$L_{FMM} = \lambda_{txt} L_{txt}(T_f, \hat{T}_f) + \lambda_{scr} L_{bce}(s, \hat{s}), \tag{2}$$

where $\hat{T}_f$ is the annotated ground truth text answer containing $< \text{SCR} >$ token.

## 4 EXPERIMENTS

### 4.1 EXPERIMENTAL PROTOCOLS

#### 4.1.1 DATASETS

To demonstrate the effectiveness of our design, we adopt the CityRefer (Miyanishi et al., 2023) dataset and a self-annotated dataset called **CityAnchor** dataset for evaluation.

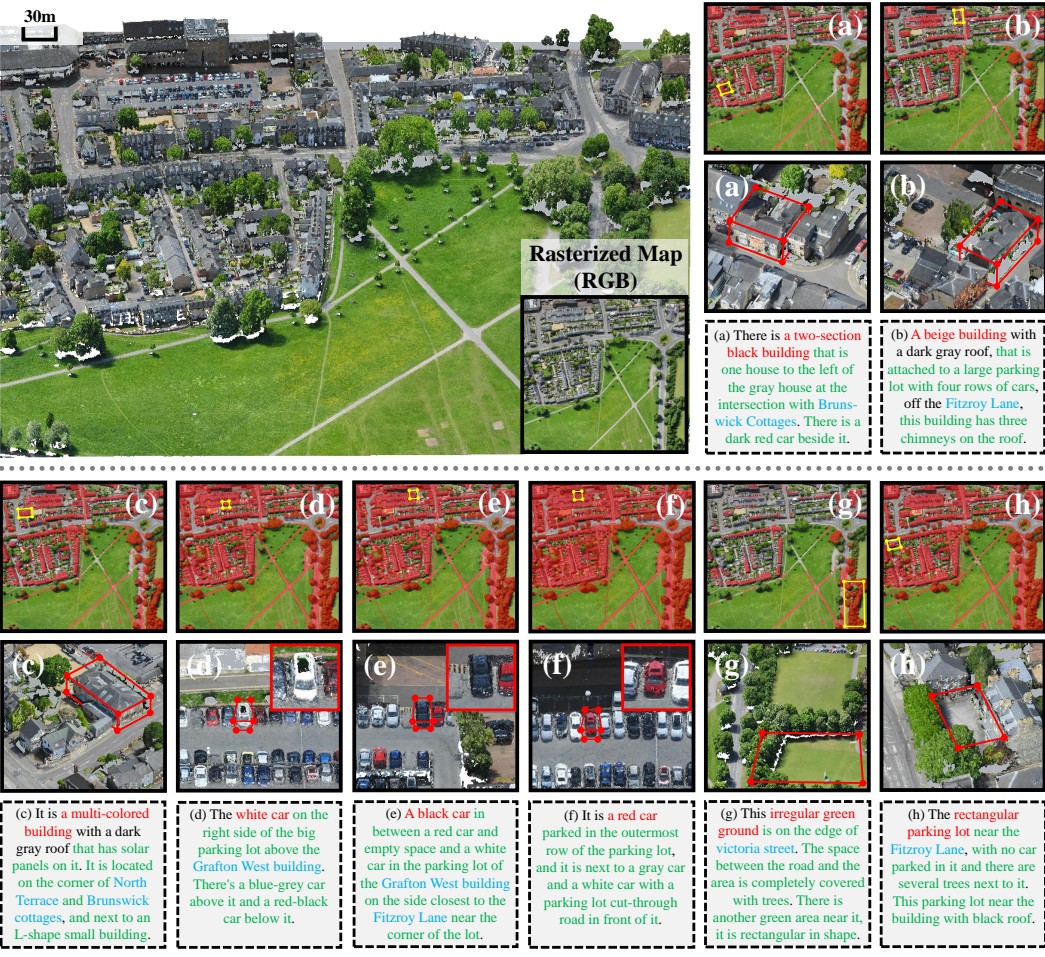

Figure 4: Qualitative results on the CityRefer (Miyanishi et al., 2023) dataset. The projected 2D map is obtained from the city-scale point cloud by top-view projection. The candidate objects from CLM are represented by red masks. In the query text, the target object is marked in red, the landmark name is marked in blue, and the neighborhood description is marked in green. Grounding results are shown in red boxes.

**CityRefer** (Miyanishi et al., 2023) is a 3D visual grounding dataset annotated from city-scale dataset SensatUrban (Hu et al., 2021b) dataset. The dataset covers an urban area of more than $6km^2$ and comprises over 35,000 natural language descriptions of 3D objects, alongside more than 5,000 landmark labels annotated using OpenStreetMap data. The grounding objects mainly fall into 4 categories: Building, Car, Ground, and Parking. We use 85% of them for training and 15% of them for evaluation.

**CityAnchor** is a city-scale 3D visual grounding dataset. We use 25 city-scale point clouds of STPLS3D (Chen et al., 2022) dataset and manually annotate them with text prompts. There are 1448 text-object pairs. 80% of these pairs are used in training while the rest are used in tests. The objects in the CityAnchor dataset cover 9 different categories including Building, Vegetation, Aircraft, Truck, Vehicle, LightPole, Fence, StreetSign and Bike.

### 4.1.2 BASELINES

We re-train and evaluate all baselines with the same settings as CityAnchor for a fair evaluation.

**InstanceRefer** (Yuan et al., 2021) is a matching-based framework designed for visual grounding on point clouds, which leverages panoptic segmentation alongside linguistic cues to identify candidate instances in point clouds.

Table 1: Quantitative results on the CityRefer (Miyanishi et al., 2023) and the CityAnchor datasets. "*NO*" and "*ND*" are abbreviations for "*Novel Objects*" and "*Novel Descriptions*", respectively.

| Method | CityRefer-NO | | CityRefer-ND | | CityAnchor-NO | | CityAnchor-ND | |
|---|---|---|---|---|---|---|---|---|
| | Acc@0.25 | Acc@0.50 | Acc@0.25 | Acc@0.50 | Acc@0.25 | Acc@0.50 | Acc@0.25 | Acc@0.50 |
| InstanceRefer | 4.09 | 3.64 | 1.93 | 1.76 | 1.58 | 1.35 | 3.04 | 2.31 |
| 3DVG-Transformer | 7.73 | 5.69 | 9.64 | 8.12 | 4.16 | 2.38 | 6.25 | 4.17 |
| EDA | 6.96 | 5.53 | 8.39 | 5.84 | 5.15 | 3.09 | 7.14 | 4.29 |
| CityRefer | 8.34 | 7.47 | 5.07 | 3.49 | 5.73 | 4.16 | 6.07 | 3.95 |
| CityAnchor | **50.69** | **46.86** | **53.17** | **50.37** | **41.23** | **35.11** | **47.81** | **43.40** |

Table 2: Additional comparisons with CityRefer (Miyanishi et al., 2023) method. "Time" means the time used in conducting visual grounding once.

| Method | CityRefer | | | | CityAnchor | | | |
|---|---|---|---|---|---|---|---|---|
| | Top-1 | Top-3 | Top-5 | Time(s) | Top-1 | Top-3 | Top-5 | Time(s) |
| CityRefer | 7.47 | 10.24 | 12.19 | 42.27 | 4.16 | 6.91 | 11.05 | 96.91 |
| CityAnchor | 46.86 | 57.12 | 59.83 | 32.45 | 35.11 | 41.98 | 48.09 | 51.72 |

**3DVG-Transformer** (Zhao et al., 2021) is a transformer-based method specifically designed for 3D visual grounding, comprehensively considering diverse relations to enhance proposal generation and facilitate cross-modality proposal disambiguation.

**EDA** (Wu et al., 2023b) is a one-step, explicit, dense-aligned approach for 3D visual grounding tasks. This approach systematically decomposes text into multiple semantic components and achieves dense alignment with corresponding visual features. It employs position-aligned and semantic-aligned loss functions to facilitate fine-grained fusion of visual-text features, thereby mitigating issues of information blurring and imbalance commonly encountered in existing methods.

**CityRefer** (Miyanishi et al., 2023) is a baseline designed for 3D visual grounding in city-scale scenes. CityRefer segments the whole scene into class-agnostic objects, compares the query text with all urban objects iteratively, and selects the most similar one as the grounding result.

### 4.1.3 METRICS AND EXPERIMENTAL SETTING

We use the Intersection over Union (IoU) between our grounding results and the ground truth objects for evaluation. We report the ratio of grounding results with an IoU larger than 0.25 and 0.5, denoted as Acc@0.25 and Acc@0.50 respectively. In evaluation, previous methods including CityRefer (Miyanishi et al., 2023) typically first select 10 candidate objects and only conduct visual grounding among these candidates. Since our target is to evaluate the performance of city-scale visual grounding, we directly apply the grounding algorithm among all hundreds of objects in a scene of around 400m×400m without manually selecting any candidates. The CityRefer dataset contains 41 scenes while the CityAnchor dataset contains 25 scenes. We evaluate the performance in two settings **"Novel Objects"** and **"Novel Descriptions"**. "Novel Objects" setting means the objects in the test set are unseen in the training set. For example, a blue car object in test set would not appear in training set. "Novel Descriptions" setting indicates that the evaluation objects are seen in the training set, but the query text descriptions are different. For example, an object may be described as "A building with white roof near the road."in training set but "Along with the street, there is a white-roofed building" in test set.

### 4.1.4 IMPLEMENTATION DETAILS

All the experiments are implemented with PyTorch on a single NVIDIA A100 GPU (40 GB). In the coarse localization stage, we obtain the input RGB map by rasterizing the top-view projection of the point cloud with a resolution of 0.1m. We use the pre-trained weights from LISA (Lai et al., 2023) to initialize our Coarse Localization Module. For our Fine-grained Matching Module, we use the weights of Vicuna-7b-v1.3 (Zheng et al., 2024) to initialize the modules inherited from LLaVA architecture. All fine-tuning is conducted with the LoRA layers (Hu et al., 2021a). We use the AdamW optimizer with a batch size of 8 and a learning rate decaying from 2e-5 to 2e-7 with a cosine annealing scheduler. The training of CLM and FMM takes about 12 and 15 hours to converge. We

set the threshold $\theta$ for the candidate object detection in CLM to a fixed 0.3 (except for the specialized analysis of RoI threshold) and the number of neighboring objects $K$ for spatial context-aware feature enhancement in FMM to 5. We select positive and negative samples in a ratio of 1:3 for FMM training.

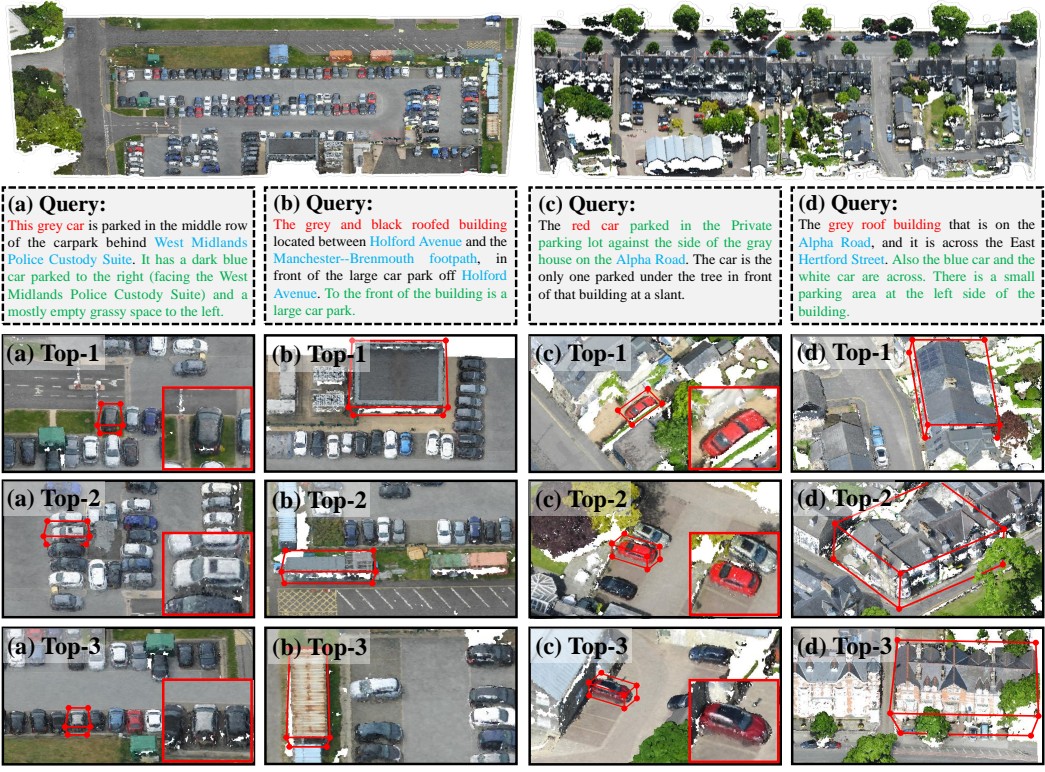

Figure 5: Top-3 objects retrieved by CityAnchor. The bounding boxes of grounding results are displayed in red. In the query text, the target object is marked in red, the landmark name is marked in blue, and the neighborhood statement is marked in green. CityAnchor distinguishes the correct target (Top-1) from quite similar candidates by considering the neighborhood information.

## 4.2 QUALITATIVE RESULTS

In Fig. 4, we provide visual grounding results of our CityAnchor on the CityRefer dataset and more qualitative results and comparisons on both datasets are provided in the Appendix. It can be seen that the first stage of CityAnchor can identify the candidate objects effectively, which provides a good initialization for efficient object matching in the second stage. In Fig. 4 (d-f), we showcase the results of cars in the same city. It can be seen that CityAnchor can accurately identify the target cars even when there are many similar candidate cars. The strong performance mainly comes from our robust object encoding that enables the LLM in CityAnchor to discriminatively match the candidate objects with the query text descriptions in terms of geometry, colors, and spatial contexts.

## 4.3 QUANTITATIVE COMPARISON

The quantitative results on the two datasets are reported in Table 1. Baselines (Yuan et al., 2021; Zhao et al., 2021; Wu et al., 2023b) manually design the network structures to learn features for 3D visual grounding, which does not perform well on the city-scale datasets containing more objects with complex spatial contexts. CityRefer (Miyanishi et al., 2023), designed for city-scale grounding, shows reasonable results on the city-scale visual grounding task. The proposed CityAnchor surpasses all baseline methods by $36\% \sim 51\%$ on Acc@0.25 and $31\% \sim 48\%$ on Acc@0.50 on two datasets, which demonstrates that our methods can effectively handle 3D visual grounding on a city scale.

Table 3: Ablation studies on the CityRefer dataset. When not using Stage 1, we iterate all the objects to find the most matched one with the text description. $E_x^s$ means the CLIP (Radford et al., 2021) feature extracted on the top-view of 3D object in FMM. $E_x^g$ means the Uni3D (Liu et al., 2024a) feature extracted on 3D object in FMM. $E_x^l$ means the encoding of the landmark names in object encoding of FMM. "Nei. Enh." means enhancing the object features with spatial context information in object encoding of FMM. "Time" means the time used in one visual grounding inference.

| Id | Stage I | Stage II | | | | Acc@0.50 | | | | | Time(s) |
|---|---|---|---|---|---|---|---|---|---|---|---|
| | | $E_x^s$ | $E_x^g$ | $E_x^l$ | Nei. Enh. | Building | Car | Ground | Parking | Overall | |
| a) | ✓ | ✓ | | | | 25.27 | 36.17 | 28.26 | 21.51 | 31.82 | - |
| b) | ✓ | ✓ | ✓ | | | 26.71 | 45.48 | 30.43 | 21.25 | 38.01 | - |
| c) | ✓ | ✓ | ✓ | ✓ | | 26.98 | 50.94 | 44.38 | 36.23 | 43.12 | - |
| d) | | ✓ | ✓ | ✓ | ✓ | 24.91 | 23.65 | 40.91 | 37.16 | 25.10 | 83.26 |
| e) | ✓ | ✓ | ✓ | ✓ | ✓ | **28.52** | **56.25** | **45.64** | **37.97** | **46.86** | 32.45 |

Table 4: Quantitative analysis of RoI thresholds of CLM using accuracy as the metrics. "Time" means the time used in conducting visual grounding once.

| RoI threshold $\theta$ | Top-1 | Top-2 | Top-3 | Top-5 | Top-10 | Time(s) |
|---|---|---|---|---|---|---|
| 0.25 | 40.92 | 45.93 | 47.27 | 50.04 | 54.13 | 48.11 |
| 0.30 | 46.86 | 53.31 | 57.12 | 59.83 | 64.66 | 32.45 |
| 0.35 | 26.80 | 28.85 | 30.48 | 32.36 | 35.10 | 27.05 |
| 0.40 | 19.66 | 22.20 | 23.05 | 24.17 | 26.39 | 21.19 |

## 4.4 MODEL ANALYSIS

**Analysis on Top-k results**. In Table 2, we further conduct a top-k experiment in comparison with CityRefer (Miyanishi et al., 2023) on the two city-scale datasets. For each query description, we select $k$ objects with top-$k$ matching scores, and $k$ is set to 1, 3, and 5. Then, we report the correctness ratio of Top-$k$ objects, where a result is regarded as correct if top-$k$ results contain an object with a larger IoU than 0.5 with the ground-truth object, i.e. Acc@0.5. It can be seen that CityAnchor consistently outperforms the baseline CityRefer by a large margin on all top-$k$ accuracy. In Fig. 5, we visualize the top-3 grounding results produced by CityAnchor. It can be seen that the top-3 results predicted by CityAnchor are all reasonable and partially aligned with the text descriptions like in colors. CityAnchor is able to distinguish them by other text descriptions about the spatial contexts of the objects. For example, in Fig. 5 (a), "a mostly empty grassy space to the left" is critical to localize the most matched car rather than the other two cars. This mainly comes from CityAnchor considering both the object properties and the spatial contexts in the second fine-grained matching stage. Also, we further report the time used by CityAnchor once on both datasets. CityAnchor requires only 32.45s to localize an object in a scene of the CityRefer dataset and is $1.3\times$ faster than the CityRefer method. This is mainly attributed to our coarse-to-fine strategy, which filters out most irrelevant objects.

**Ablation Studies**. In Table 3, we conduct ablation studies of our designs and report Acc@0.50 on the CityRefer dataset. We validate the effectiveness of the candidate object detection in CLM and the feature extraction in FMM. Comparing a) and b), we observe that encoding 3D object features $E_x^g$ using Uni3D (Liu et al., 2024a) and the projected 2D features $E_x^s$ using CLIP (Radford et al., 2021) both improve the grounding performance. Comparison between b) and c) highlights that encoding landmark names $E_x^l$ into object features enables FMM to achieve more accurate grounding results, particularly for targets strongly related to landmarks such as grounds and parking. Comparison between d) with e) reveals that utilizing the predicted candidate object selection of our CLM accelerates inference by $2.5\times$ and significantly improves the grounding accuracy by $\sim 21\%$. Comparing c) with e) demonstrates that incorporating spatial context information from neighboring objects improves the accuracy of localization in the grounding task.

**Analysis on ROI threshold**. In the first stage, we use a threshold $\theta$ to determine the RoI output by CLM, and grounding is only conducted on objects within the RoI. We perform an analysis on the threshold using grounding accuracy and runtime as metrics. As shown in Table 4, a low threshold results in numerous candidate objects being involved in fine-grained comparisons, thereby leading to long grounding times. Conversely, setting a stricter threshold results in the exclusion of correct target

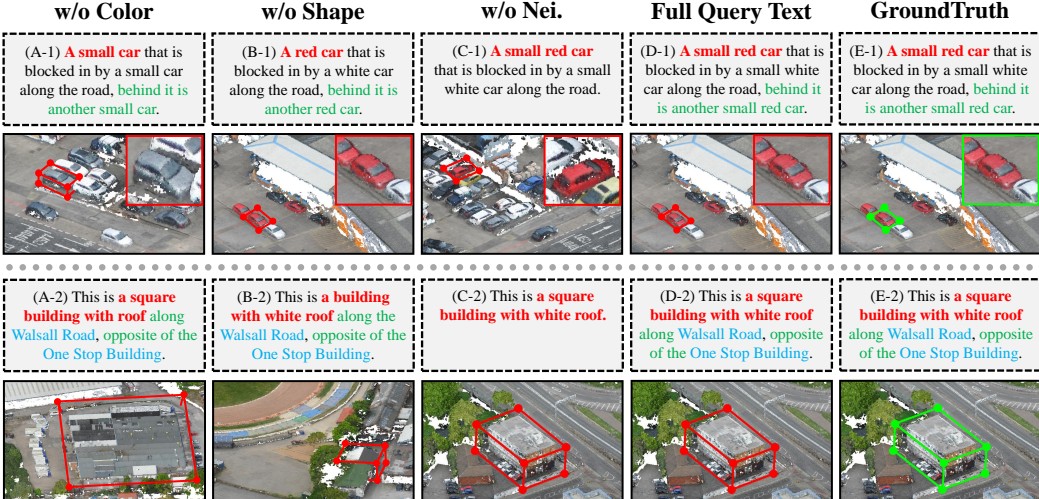

Figure 6: Analysis for specific parts of text prompts on two representative examples. "w/o Color", "w/o Shape" and "w/o Nei." mean removing the description related to color, shape and neighborhood information in query text, respectively.

objects, leading to a decrease in grounding accuracy. To strike a balance between grounding accuracy and time consumption, we selected 0.3 as the RoI threshold for the CLM in the first stage.

**Analysis on different text descriptions**. To show the compatibility with different styles of text prompts, we perform extended experiments to evaluate the impact of different text descriptions on grounding performance. As shown in Fig. 6, we visualize the grounding results of systematically removing text descriptions related to color, shape, and neighborhood contextual information. Color description plays an important role in visual grounding and the lack of color information often leads to incorrect results. In contrast, while shape descriptions are less critical for cars, they become more significant for buildings due to the larger variability of shapes. In a scene with many similar objects (e.g., red cars), color and shape descriptions are inadequate to distinguish these objects, and integrating neighborhood contextual information is vital for achieving accurate grounding.

## 5 LIMITATIONS AND CONCLUSION

**Limitations**. Though our method produces reasonable results on the city-scale 3D visual grounding task, there are still limitations. The inference time is still long, which costs ~32s for a text description. This inference process could be sped up by quantization and pruning as done in other LLM applications. We can incorporate landmark information into CityAnchor as external knowledge using Retrieval-Augmented Generation (RAG) (Gao et al., 2023), utilizing landmark information for efficient grounding. Another limitation is that we need to adjust the RoI threshold during the coarse localization stage, as the sizes and shapes of objects differ on the projected RGB map. Our future work will involve improving the efficiency and generalizability of RoI detection as well as extending CityAnchor to conduct grounding for complex and dynamic 3D city-scale point clouds.

**Conclusion**. We present CityAnchor, a 3D visual grounding method designed for city-scale scenes. The key component of CityAnchor is a two-stage multi-modality LLM. In the coarse stage, CityAnchor predicts candidate objects that correspond to the query text description on the projected 2D maps from the urban point cloud, which enables us to quickly rule out irrelevant regions and focus on plausible objects. In the fine stage, CityAnchor conducts fine-grained matching between the text descriptions and the possible objects in the above candidate objects to determine the final grounding results by encoding both text descriptions and the features of objects into an LLM. We manually annotate a new dataset called CityAnchor and evaluate our method on both the CityAnchor dataset and the CityRefer dataset. The results show that our method can produce accurate visual grounding on city-scale point clouds.

## 6 ACKNOWLEDGEMENT

This research is jointly sponsored by Projects of the National Key Research and Development Program of the Ministry of Science and Technology, China 2022YFB3904102 and NSFC Basic Research Project for Ph.D. Candidate (No.424B2012).

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

# 7 APPENDIX

In the appendix section, to further demonstrate the effectiveness of our proposed city-scale visual grounding system CityAnchor, we present additional visualization of grounding results. Moreover, we introduce the additional implementation details of CityAnchor and the self-annotated CityAnchor dataset.

## A.1 ADDITIONAL QUALITATIVE RESULTS FOR CITY-SCALE VISUAL GROUNDING

In Fig A.1, we provide additional qualitative results of our method on the self-annotated CityAnchor dataset in "Novel Objects" setting. In Fig. A.2, we provide additional qualitative results of our method on the CityRefer dataset in "Novel Objects" setting. The filter effectiveness of RoI output by CLM can vary depending on the size and attribute of target objects. Large objects (e.g., factory, athletic field, parking lots) can be easily identified and excluded through the heat map output by CLM. In contrast, small objects (e.g., cars, residential buildings) are more challenging to distinguish, requiring

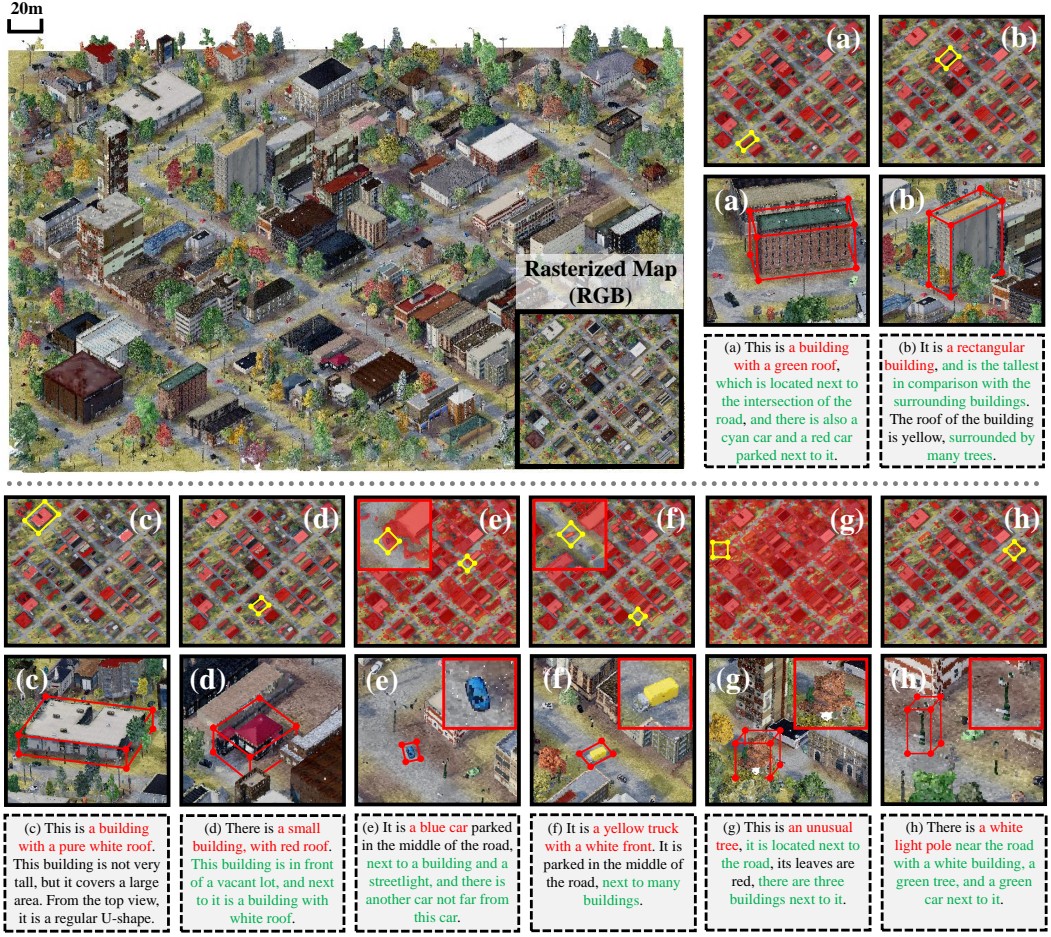

Figure A.1: Qualitative results on the CityAnchor dataset in "Novel Objects" setting. The projected 2D map is obtained from the city-scale point cloud by top-view projection. The candidate objects from CLM are represented by red masks. In the query text, the target object is marked in red, the landmark name is marked in blue, and the neighborhood statement is marked in green. Grounding results are shown in red boxes.

further detailed comparison in the FMM. In Fig. A.3, we provide qualitative comparisons between the proposed CityAnchor and the baseline method CityRefer, where CityAnchor can distinguish the correct target for similar erroneous objects by considering both the object attributes and the spatial context information. In Fig A.4 and Fig A.5, we provide additional qualitative results of our method on the CityRefer dataset in "Novel Descriptions" setting.

We have several observations from them. First, color information is frequently mentioned in each query. Encoding the RGB information of an object with Uni3D and CLIP is particularly important for distinguishing similar vehicles or buildings, as the color of the car or roof can serve as the most immediate and intuitive visual cue for object description. In addition, landmark information is also indispensable for open-world navigation and localization. Roads and parking lots are typically feature-less but have landmark names. Therefore, incorporating landmarks into CityAnchor promotes the ability to focus on specific objects through their proprietary names, thereby improving the accuracy of 3D visual grounding.

## A.2 EVALUATION OF CANDIDATE OBJECT SELECTION IN CLM

In this section, we aim to evaluate the accuracy of the candidate objects selected by the coarse localization module. Here, we adopt the same threshold to segment the heatmap. With this threshold,

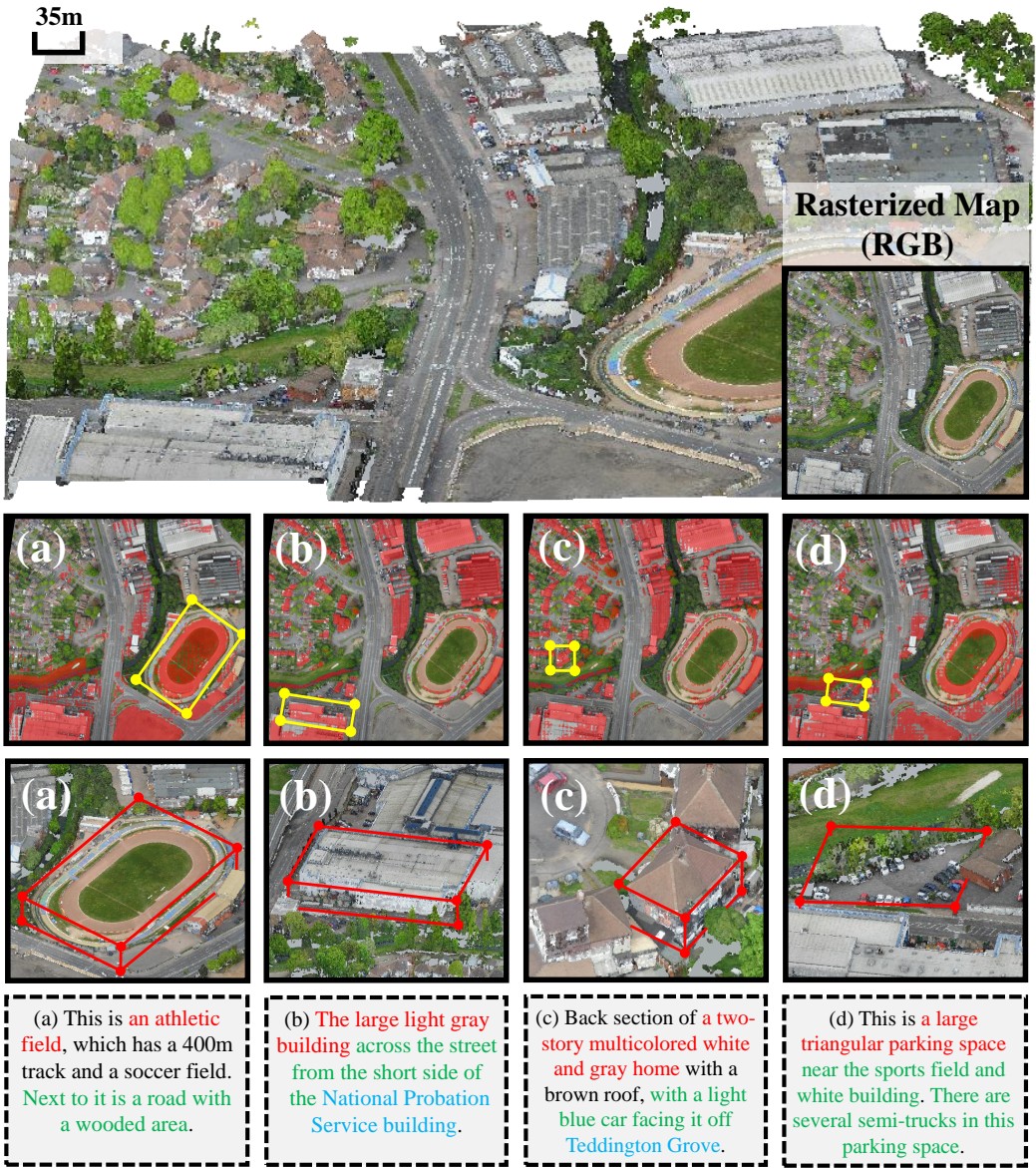

Figure A.2: Additional Qualitative results on the CityRefer dataset in "Novel Objects" setting. The projected 2D map is obtained from the city-scale point cloud by top-view projection. The candidate objects from CLM are represented by red masks. In the query text, the target object is marked in red, the landmark name is marked in blue, and the neighborhood statement is marked in green. Grounding results are shown in red boxes.

we turn it into a binary heatmap and evaluate its IoU with the region of the ground-truth object on this 2D map. We call these activation regions on the heatmap as Region of Interests (RoI).

### A.2.1 MODEL TRAINING

The training data for the CLM is derived exclusively from CityRefer (Miyanishi et al., 2023) dataset and the CityAnchor dataset. We project the point cloud instances onto the XoY plane to obtain rasterized object masks and then densify them using traditional erosion and dilation operations. Finally, we feed the text-image-mask pairs into the CLM for training and validation. During the training process, the weights of the text generation loss $\lambda_{txt}$ and the mask loss $\lambda_{seg}$ are set to 1.0 and 1.0, respectively. Besides, we set the batch size as 2 and the number of epochs as 5.

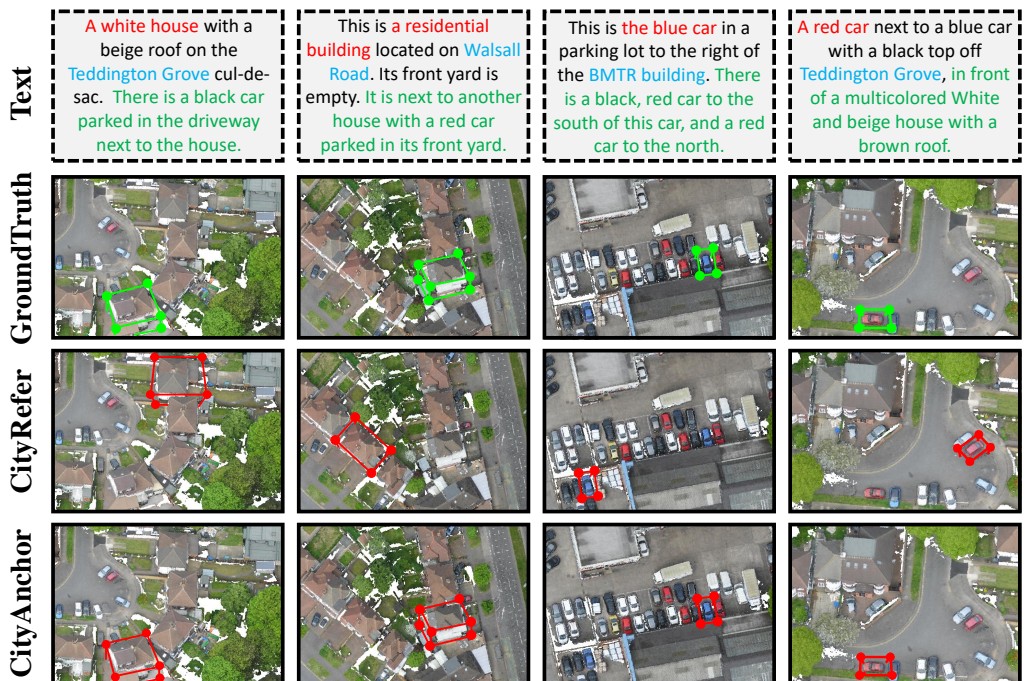

Figure A.3: Qualitative comparisons of the baseline method CityRefer and the proposed framework CityAnchor. The ground truth and predicted boxes are displayed in green and red, respectively.

Table A.1: Quantitative analysis of RoI area segmentation on CityRefer and CityAnchor datasets.

| Method | CityRefer | | CityAnchor | |
|--------|-----------|------|------------|------|
| | gIoU | cIoU | gIoU | cIoU |
| LISA-7B | 11.5 | 10.1 | 13.4 | 13.0 |
| LISA-13B | 16.2 | 13.7 | 17.1 | 14.9 |

### A.2.2 EVALUATION METRICS

Following most previous works (Kazemzadeh et al., 2014; Mao et al., 2016) on image segmentation, we adopt two metrics: gIoU and cIoU. The gIoU is defined as the average of all per-image Intersection-over-Unions (IoUs), while cIoU is determined by the cumulative intersection over the union. Given that cIoU tends to be highly biased toward large-range objects and exhibits significant fluctuations, gIoU is the preferred metric.

### A.2.3 CANDIDATE OBJECT SELECTION RESULTS

**Visualization**. As illustrated in Fig A.6, We further visualize the heat maps, on which we have performed smoothing. The value of each pixel in the heatmap indicates the correlation degree with the text description (red color indicates a strong correlation and blue color indicates a weak correlation).

**Quantitative analysis**. As shown in Table A.1, we show the IoU of RoI on two datasets after model fine-tuning using different LLM backbones. In this RoI regression task from a projected 2D map, LISA-13B (Lai et al., 2023) demonstrates superior performance compared to LISA-7B (Lai et al., 2023) on both the Cityrefer and Cityanchor datasets, particularly when the query text descriptions are long, which indicates that understanding text descriptions and extracting discriminative features remain performance bottlenecks. A more powerful LLM backbone could improve performance in the segmentation task.

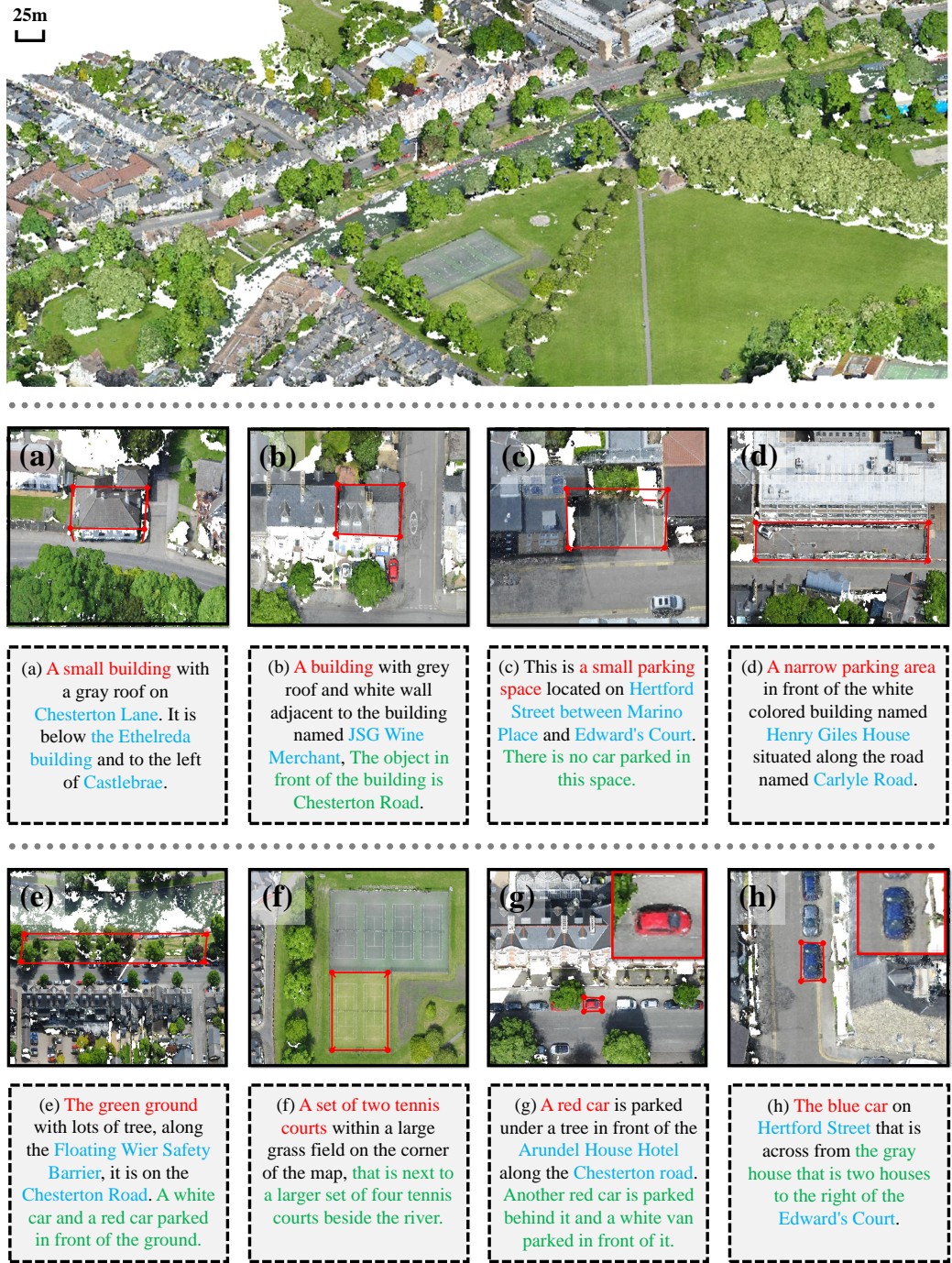

Figure A.4: Additional qualitative results of Scene I on the CityRefer dataset in "Novel Descriptions" setting. The bounding boxes of resulting objects are drawn in red. In the query texts, the target object is marked in red, the landmark name is marked in blue, and the neighborhood statement is marked in green.

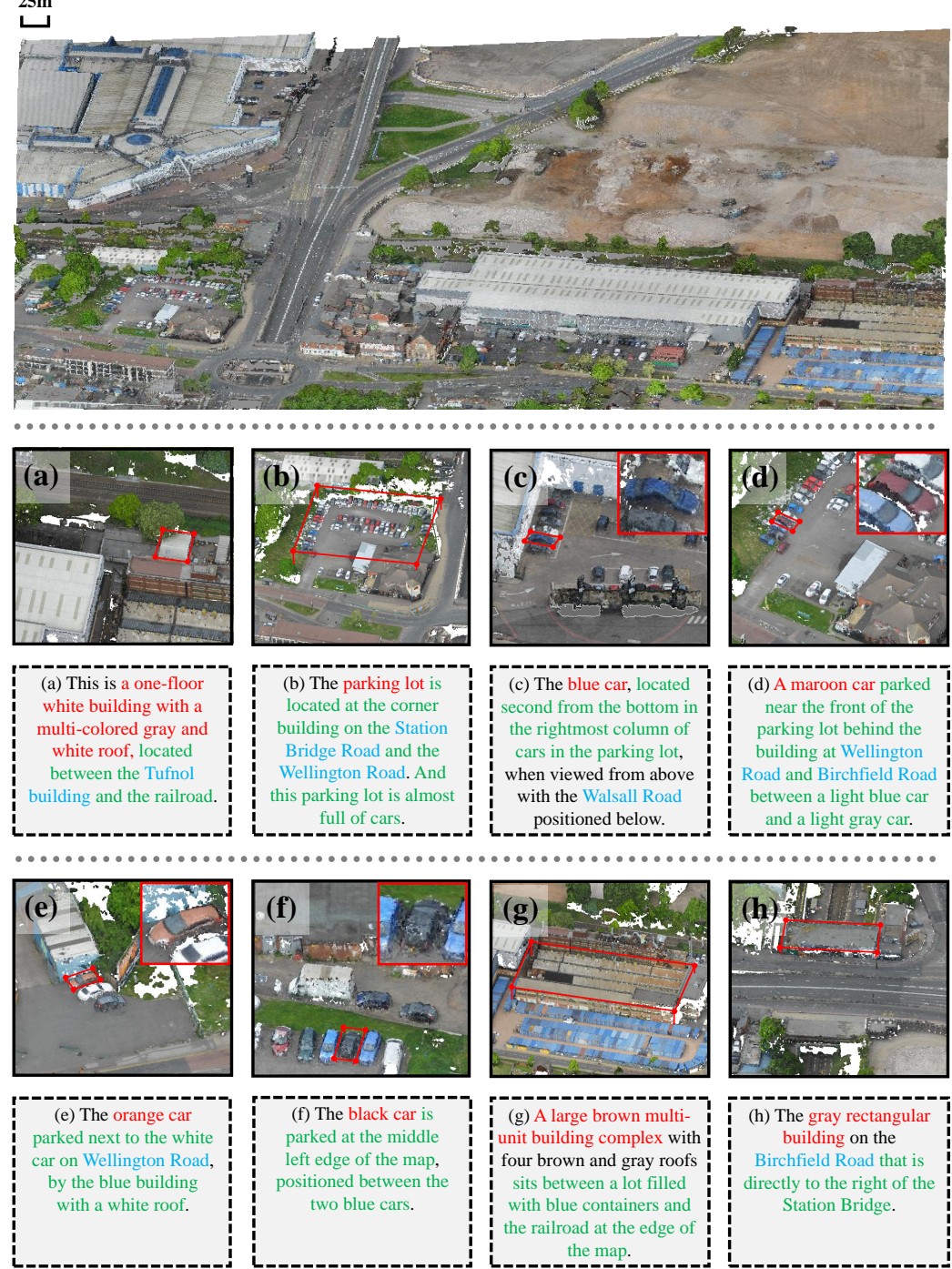

Figure A.5: Additional qualitative results of Scene II on the CityRefer dataset in "Novel Descriptions" setting. The bounding boxes of resulting objects are drawn in red. In the query texts, the target object is marked in red, the landmark name is marked in blue, and the neighborhood statement is marked in green.

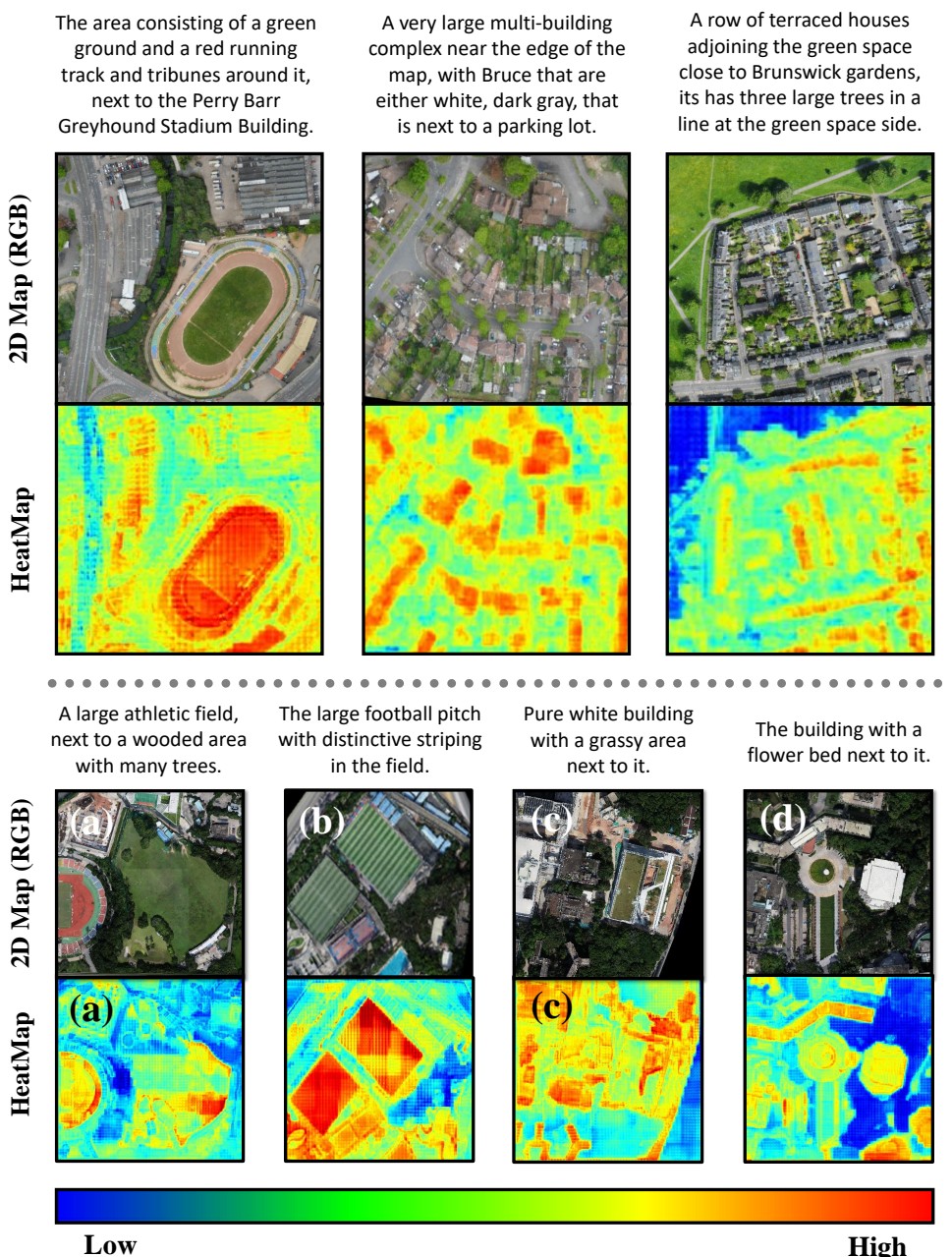

Figure A.6: Visualisation of predicted heat maps.

Table A.2: 3D instance segmentation results on CityAnchor (STPLS3D) dataset.

| Target | AP | $AP_{50}$ | $AP_{25}$ | mRec | $mRec_{50}$ | $mRec_{25}$ |
|---|---|---|---|---|---|---|
| Ground | 74.5 | 77.7 | 79.5 | 75.4 | 78.2 | 79.8 |
| Vegetation | 32.8 | 58.3 | 71.1 | 43.9 | 68.9 | 80.5 |
| Building | 32.4 | 48.9 | 59.4 | 41.2 | 60.7 | 76.9 |
| Wall | 27.2 | 30.8 | 34.5 | 28.0 | 31.4 | 35.7 |
| Bridge | 84.2 | 94.7 | 96.7 | 85.7 | 95.5 | 97.6 |
| Parking | 78.3 | 88.6 | 91.6 | 82.0 | 92.4 | 96.7 |
| Rail | 58.6 | 92.0 | 92.2 | 64.6 | 93.8 | 96.9 |
| Traffic Road | 72.2 | 80.2 | 82.9 | 86.8 | 94.2 | 97.1 |
| Street Furniture | 18.0 | 38.4 | 52.5 | 34.2 | 60.9 | 79.2 |
| Car | 54.9 | 79.0 | 88.0 | 61.3 | 83.0 | 91.7 |
| Footpath | 48.5 | 68.0 | 74.6 | 53.8 | 70.7 | 76.7 |
| Bike | 13.2 | 25.6 | 38.6 | 17.4 | 30.3 | 47.6 |
| Water | 48.2 | 61.5 | 66.9 | 58.0 | 73.7 | 84.5 |
| Fence | 19.4 | 40.1 | 71.6 | 26.2 | 50.5 | 78.8 |
| Average | 47.3 | 63.1 | 71.4 | 54.2 | 70.3 | 80.0 |

## A.3 3D INSTANCE SEGMENTATION

In this section, we conduct an experiment on the STPLS3D (Chen et al., 2022) dataset to evaluate the model performance of 3D instance segmentation on point clouds. The primary objective is to accurately generate segmentation masks, each associated with a specific category label, for each identifiable object.

### A.3.1 ARCHITECTURE

We use SoftGroup (Vu et al., 2022) as the primary network for our city-scale instance segmentation task on the STPLS3D dataset. The methodology is split into two stages: bottom-up grouping and top-down refinement.

### A.3.2 NETWORK TRAINING

3D instance segmentation in city-level dataset was developed using the PyTorch deep learning framework. The SoftGroup model was trained on 10,000 iterations with Adam optimizer (Kingma & Ba, 2014). The batch size was maintained at 32, and the initial learning rate was set at 0.05, modulated via a cosine annealing schedule. The parameters such as voxel size and grouping bandwidth were configured at 1.0 meters and 2.0 meters, respectively.

### A.3.3 EVALUATION METRICS

The evaluation metric for instance segmentation model performance is the standard average precision, we utilized average precision (AP) and mean recall (mRec) as the primary evaluation metrics for each class assessed. $AP_{50}$ and $AP_{25}$ denote the scores with IoU thresholds of 50% and 25%, respectively. AP denotes the averaged scores with IoU threshold from 50% to 95% with a step size of 5%.

### A.3.4 RESULTS ON STPLS3D DATASET

Table A.2 shows the 3D instance segmentation performances. In the STPLS3D dataset, the average $AP_{50}$ of the SoftGroup method achieves 47.3%.

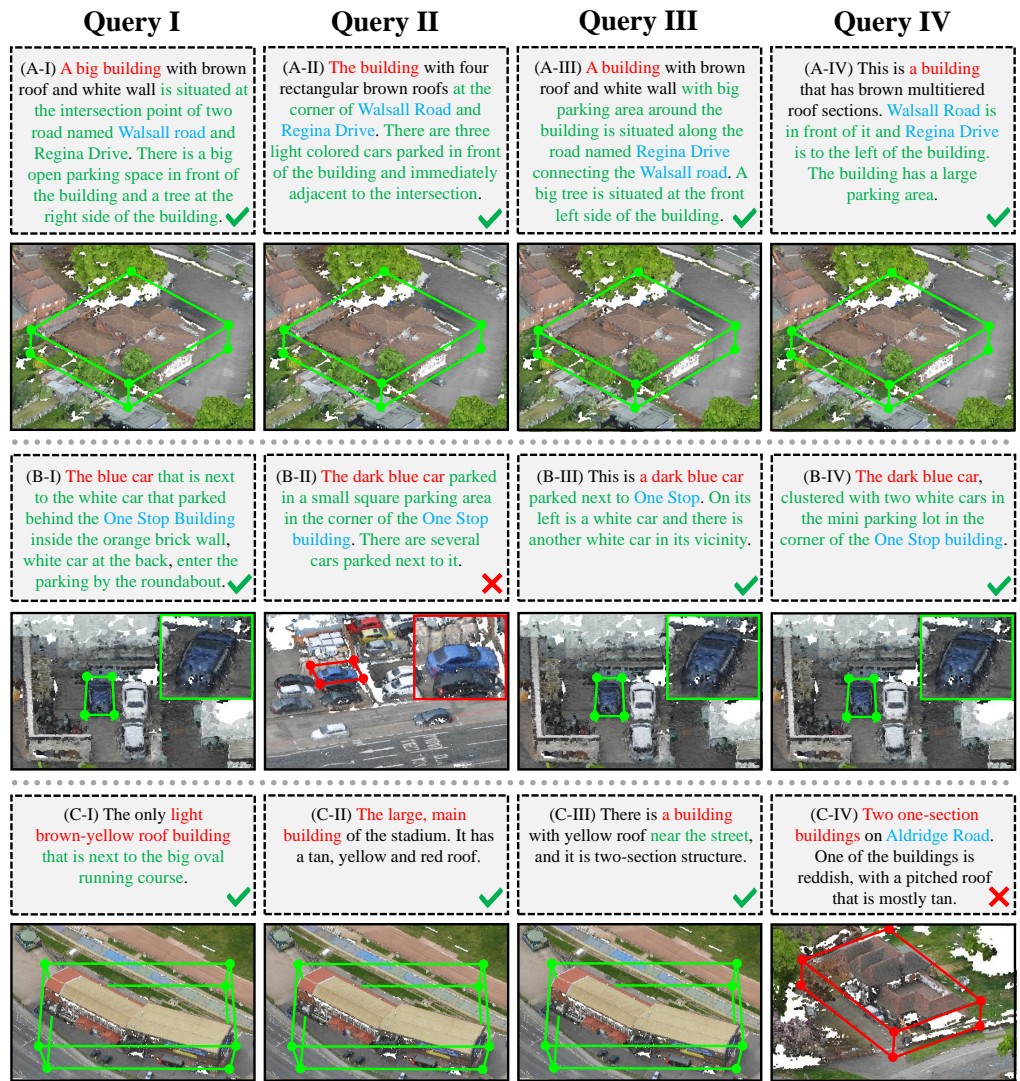

Figure A.7: Analysis for diverse text prompts toward same object on three representative examples. Note that a green checkmark in the bottom right corner of the query text box indicates a correct grounding case, whereas a red cross indicates an incorrect grounding case.

## A.4 ANALYSIS FOR DIVERSE TEXT PROMPTS TOWARD SAME OBJECT

In Fig. A.7, we provide qualitative results for diverse text prompts toward same object. In addition to thorough object feature extraction, the text description plays a crucial role in visual grounding. For the successful grounding cases, text descriptions related color and category are often indispensable.

## A.5 GENERALITY EXPERIMENT FOR UNKNOWN OBJECTS

As illustrated in Fig A.8, we conduct a generality experiment to assess CityAnchor's ability to recognize unknown objects. In CityRefer dataset, CityAnchor is trained with only four object categories: Building, Car, Parking, and Ground, and we introduce unknown object categories (such as Road Intersection, Woodland and River) that CityAnchor has not seen before. We evaluate the CityAnchor's ability to predict the similarity between these unknown objects and both the correct text descriptions (Positive Sample) and incorrect text descriptions (Negative Sample). In Fig. A.9, we provide both the RoI map and grounding results for three representative unknown objects.

Given the Fine-grained Matching Module (FMM) in CityAnchor is fine-tuned from LLaVA, which possesses outstanding real-world recognition capabilities, CityAnchor effectively inherits these

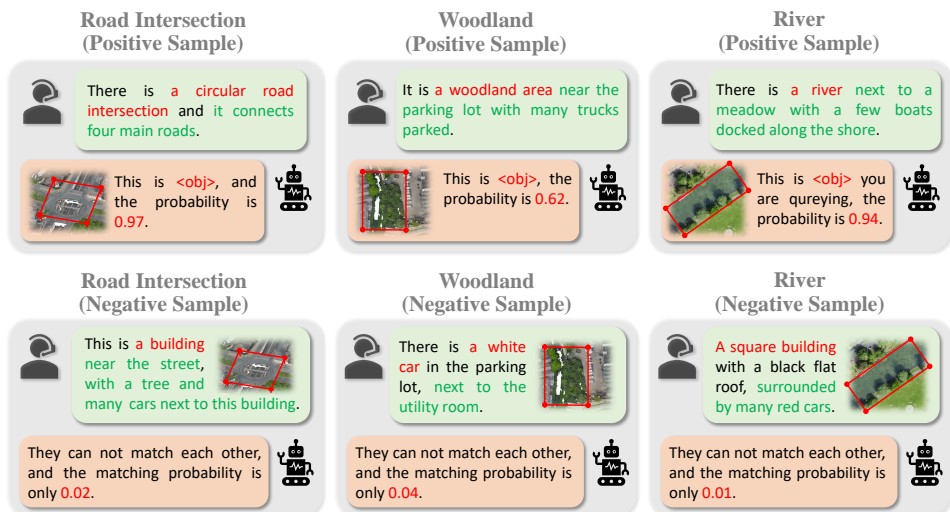

Figure A.8: Qualitative results on unknown objects for pre-trained CityAnchor. The positive sample represents the case where the object matches the query text, with the matching probability of 1 in GroundTruth, while the negative sample represents the mismatched case, with the matching probability of 0 in GroundTruth.

Table A.3: Comparison of existing 3D visual grounding datasets in outdoor environment. $N_d$ means the number of text descriptions for 3D objects. $N_{points}$ means the number of points. UAV-P means Unmanned Aerial Vehicle (UAV) Photogrammetry. SA-P means Synthetic Aerial (SA) Photogrammetry.

| Dataset | $N_d$ | Area | Source | $N_{points}$ |
|---|---|---|---|---|
| TouchDown | 25,575 | Roadside | Google Street View | - |
| KITTI360Pose | 43,381 | Roadside | KITTI360 (3D Scan) | 1,000M |
| CityRefer | 35,196 | City Center | SensatUrban (UAV-P) | 2,847M |
| CityAnchor | 1448 | City | STPLS3D (UAV-P + SA-P) | - |

strengths when interpreting each candidate object. Furthermore, the query texts often reference surrounding objects (e.g., roads, rivers, trees, and woodlands) to facilitate successful grounding by providing neighborhood contextual information. This requires CityAnchor to comprehend a broader range of objects and their fundamental characteristics, even when these objects are not the target objects in CityRefer dataset. The grounding results for unknown objects demonstrate that CityAnchor exhibits the certain generalization capabilities, enabling it to perform 3D visual grounding across a broader range of objects.

## A.6 CITYANCHOR DATASET

As shown in Table A.3, we provide the basic statistics of the CityAnchor dataset in comparison to TouchDown (Chen et al., 2019), KITTI360Pose (Kolmet et al., 2022) and CityRefer (Miyanishi et al., 2023) datasets.

The TouchDown dataset is derived from open-access Google Street View, aiming at text-guided navigation and spatial reasoning using real-life visual observations. KITTI360Pose dataset is oriented towards outdoor traffic environments and specifically designed to enhance mobile robot localization using templated language descriptions that focus solely on positional data. However, both the TouchDown and KITTI360Pose datasets are derived from a vehicle-based system, which is constrained to roadside environments. Although the benchmark dataset CityRefer is constructed on the basis of a publicly available 2D map (OpenStreetMap), visual grounding in general may result in privacy issues or racial and gender biases. To this end, we propose CityAnchor dataset, which is a synthesized city-scale 3D visual grounding benchmark, we annotated using 25 city-scale point clouds

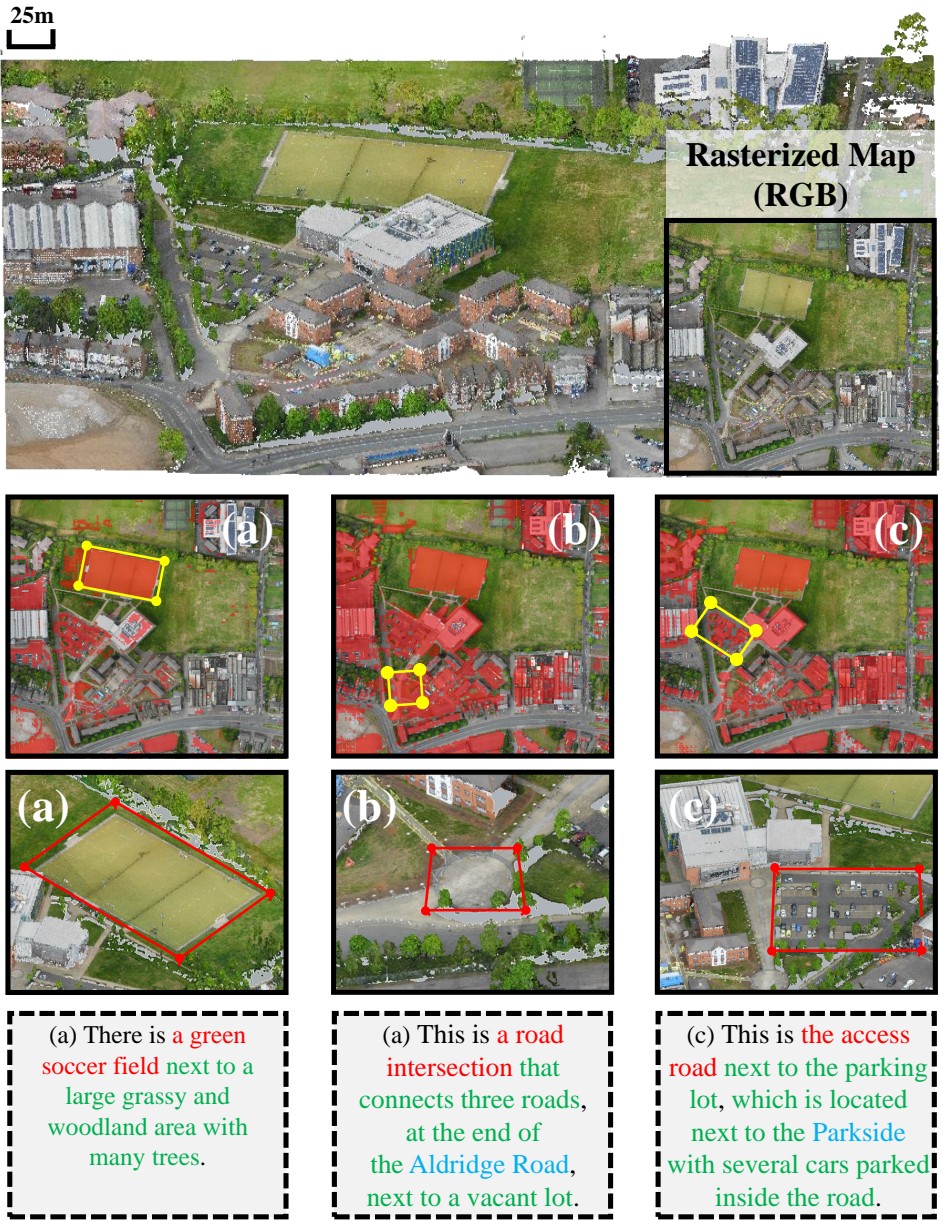

Figure A.9: Qualitative results for unknown objects on the CityRefer dataset. The bounding boxes of resulting objects are drawn in red. In the query texts, the target object is marked in red, the landmark name is marked in blue, and the neighborhood statement is marked in green.

of STPLS3D (Chen et al., 2022) dataset. Based on the annotated categories, we add text descriptions including specific numerical information (e.g. floor heights, coverage areas, distances, etc.).

### A.7 POSSIBLE BROADER IMPACTS

City-scale visual grounding is not well-studied in the literature. This technique may leak some privacy data to enable a person to localize some sensitive targets. This could be addressed in the data preparation to prevent this from appearing in the annotations.

### A.8 LICENSE OF DATASETS

The CityRefer dataset is under MIT license while the STPLS3D dataset is under CC-BY-NC-SA 4.0 license. We will release our CityAnchor dataset under the MIT license.

