# OpenReview forum: "CityAnchor: City-scale 3D Visual Grounding with Multi-modality LLMs"
_ICLR.cc/2025/Conference — ICLR 2025 Poster_

### Official Review · Reviewer_P5JV · 2024-10-30

**Soundness:** 3
**Presentation:** 3
**Contribution:** 2
**Rating:** 6
**Confidence:** 4

**Summary:**

The paper presents CityAnchor, a novel 3D visual grounding method designed for city-scale scenes, utilizing a two-stage multi-modality large language model (LLM). The first stage focuses on coarse localization of potential object regions based on text descriptions, while the second stage involves fine-grained matching to determine the best object matches. Specifically,
- Coarse Localization Module (CLM)  takes a city-scale colored point cloud, projects it onto a 2D map, and uses the text description to regress possible regions of the target on this 2D map.
- Fine-grained Matching Module (FMM) performs fine-grained comparisons between each candidate object identified by the CLM and the text description by predictiong the similarity between the text and each candidate object, and select the object with the highest similarity score.
Additionally, the paper introduces the CityAnchor dataset, a synthesized benchmark for 3D visual grounding. The method demonstrates significant improvements in accuracy and efficiency over existing techniques on the CityRefer and CityAnchor datasets.

**Strengths:**

+ The paper introduces CityAnchor, a novel approach to 3D visual grounding specifically tailored for city-scale scenes. Its use of a two-stage multi-modality large language model (LLM) for both coarse localization and fine-grained matching is effective and advance in the field. The proposed approach effectively integrates spatial context and object features, showcases the ability in addressing the complexities of urban environments, setting it apart from existing methods that may not adequately handle such challenges.
+ The paper provides a comprehensive evaluation of the model's performance, demonstrating significant improvements in accuracy compared to existing methods. Additionally, the introduction of the CityAnchor dataset as a synthesized benchmark for 3D visual grounding further enhances the quality of the research, providing a valuable resource for future studies.
+ The paper is well-structured and easy to follow, clearly stating the objectives, methodology, and results of the research.

**Weaknesses:**

- Limitations of 2D Mapping: Projecting 3D space onto a 2D map often results in a loss of spatial information, particularly in cases of occlusions or overlapping instances. This pipeline, especially the CLM component, operates under the significant assumption that all objects are nearly flat, which inherently limits its applicability to more complex environments, such as indoor spaces or densely populated urban areas like NYC.

- Dependence on Point Segmentation Model: The pipeline requires users to first segment the scene point cloud into objects using a pretrained 3D segmentation model. As a result, it heavily depends on the granularity and accuracy of this pretrained model, setting an upper performance limit for the proposed approach.

- Justification for the Proposed Dataset: While introducing a new dataset is valuable, the necessity of the CityAnchor dataset for this task or the proposed method requires further clarification. It is essential to address whether the method’s performance would significantly degrade without this dataset, whether it enhances diversity or robustness, and if it offers any unique attributes that the CityRefer dataset lacks. Providing these details would strengthen the contribution of the dataset to the paper.

- Inference time and hyperparameter tuning: These two limitations were briefly discussed in the paper.

**Questions:**

- Effectiveness of CLM: In Figure 4, it appears that CLM is not highly effective, as it predominantly highlights houses and roads. Could the authors provide additional results or offer an explanation for this behavior?

- Prompt Engineering Requirements: It seems that users must invest effort in engineering their prompts to achieve accurate results, which may pose usability challenges. The authors could consider conducting a user study to gather prompts from a diverse set of users to assess the method’s robustness when different descriptions target the same object.

- Progressive Prompting Approach: It would be intuitive to structure prompts in a coarse-to-fine manner—beginning with a broad query that identifies multiple potential candidates and gradually introducing constraints to precisely locate the target.

---

> ### Author Response · Authors · 2024-11-21
> **Responses to Reviewer P5JV (Part 1)**
>
> **We sincerely appreciate the reviewer's insightful comments and time dedicated to evaluating our work. We address all concerns below.**
>
> **Q1:** Limitations of 2D Mapping: Projecting 3D space onto a 2D map often results in a loss of spatial information, particularly in cases of occlusions or overlapping instances. This pipeline, especially the CLM component, operates under the significant assumption that all objects are nearly flat, which inherently limits its applicability to more complex environments, such as indoor spaces or densely populated urban areas like NYC.
>
> **A1:** Thanks for pointing out the limitations of 2D mapping. We agree that the proposed method using 2D projection works in most cities similar to our two datasets but not in extremely complex 3D cities like NYC. Current existing 3D visual grounding only focuses on small-scale scenes like a room while our method CityAnchor extends our baseline methods to real city-scale visual grounding, which is a substantial step toward intelligent geospatial analysis. We agree that generalizing such visual grounding to complex 3D cities like NYC could be an important and interesting future topic. Though the current CLM may not be applicable to such complex 3D cities, incorporating an LLM like our FMM or a more generalized CLM is still a feasible and promising starting point for these complex 3D cities. We have added these discussions in the revision of **Sec.5 (Limitations)**.
>
> **Q2:** Dependence on Point Segmentation Model: The pipeline requires users to first segment the scene point cloud into objects using a pretrained 3D segmentation model. As a result, it heavily depends on the granularity and accuracy of this pretrained model, setting an upper performance limit for the proposed approach.
>
> **A2:** Thanks for pointing out the CityAnchor's dependence on a pretrained 3D segmentation model. Below, we provide a detailed response to address this concern.
>
> **a) CityAnchor has the flexibility in selecting 3D segmentation models.**
> CityAnchor follows the CityRefer to segment the scene point cloud into objects using a pretrained 3D segmentation model but can be incorporated with different instance segmentation methods. We do not rely on any specific type of instance segmentation algorithms.
>
> **b) CityAnchor has the potential to generalize to unknown objects.**
> In the revision, we additionally conduct a generalization ability experiment to assess CityAnchor's ability to recognize unknown objects in Figure 8 of the Appendix. On the CityRefer dataset, CityAnchor is trained with only four object categories: Building, Car, Parking, and Ground, and we introduce three objects of unknown object categories (Road Intersection, Woodland, and River) that CityAnchor has not seen before. We show that CityAnchor is still able to determine a reasonable matching score between the text descriptions and the given 3D objects due to the strong generalization ability of LLMs, even though these objects may not be well segmented by the 3D segmentation model.
>
> **Q3:** Justification for the Proposed Dataset: While introducing a new dataset is valuable, the necessity of the CityAnchor dataset for this task or the proposed method requires further clarification. It is essential to address whether the method’s performance would significantly degrade without this dataset, whether it enhances diversity or robustness, and if it offers any unique attributes that the CityRefer dataset lacks. Providing these details would strengthen the contribution of the dataset to the paper.
>
> **A3:** Thanks for your comments.
> We utilize the CityAnchor dataset to illustrate the effective design of our method because **1)** the CityAnchor dataset has a wider range of object categories, covering nine categories including Building, Vegetation, Aircraft, Truck, Vehicle, LightPole, Fence, StreetSign, and Bike, which demonstrates the effectiveness of the grounding algorithm on more categories. **2)** the CityAnchor dataset is based on synthetic data and thus has accurate instance labels, which helps us evaluate the visual grounding more accurately without other distracting factors. Thus, we have provided this dataset for the evaluation of our visual grounding performance.

---

> ### Author Response · Authors · 2024-11-21
> **Responses to Reviewer P5JV (Part 2)**
>
> **Q4:** Inference time and hyperparameter tuning: These two limitations were briefly discussed in the paper.
>
> **A4:** Thanks for your comments. We have quantitatively discussed inference time and hyperparameter tuning for RoI threshold as much detail as possible in the main paper. Below, we provide a detailed response to clarify these points.
>
> **a) Inference time.** We have discussed the inference time in Table 2 and Table 3 of the main paper. CityAnchor takes only $32.45s$ and $51.72s$ to localize an object in a scene of the CityRefer and CityAnchor dataset, which is $1.3\times$ and $1.8\times$ faster than the CityRefer method, respectively. This is mainly attributed to our coarse-to-fine strategy for filtering out most irrelevant objects. Additionally, utilizing the CLM in CityAnchor accelerates inference time by $2.5 \times$ compared to the configuration without CLM.
>
> **b) RoI threshold in hyperparameter tuning.**
> As shown in Table 4 of the main paper, we have discussed the RoI threshold in hyperparameter tuning. A low threshold results in numerous candidate objects being involved in fine-grained comparisons, thereby leading to long grounding times. Conversely, setting a stricter threshold results in the exclusion of correct target objects, leading to a decrease in grounding accuracy. To strike a balance between grounding accuracy and time consumption, we select $0.3$ as the RoI threshold for the CLM in the first stage.
>
> From the above analysis, our CityAnchor is faster and better than all the baseline methods, because the CLM effectively filters out erroneous objects.
>
> **Q5:** Effectiveness of CLM: In Figure 4, it appears that CLM is not highly effective, as it predominantly highlights houses and roads. Could the authors provide additional results or offer an explanation for this behavior?
>
> **A5:** Thanks for your comments. The rasterized map shown in Figure 4 is mainly dominated by small objects, which shows the limited effectiveness of CLM in this case.
> As illustrated in Figure 2 of the Appendix, we provide additional qualitative results of our method on the CityRefer dataset in "Novel Objects" setting, which shows that CLM can effectively filter our erroneous objects.
>
> As illustrated in Figure 6 of the Appendix, we provide heat map visualizations for both simple and complex query texts, which shows that CLM performs reasonably even on complex texts.
> We have shown in Table 3 of the main paper that CLM significantly reduces computation time while improving grounding accuracy, with 46.86\% of Acc@0.50 and only $32.45s$ in localizing an object.
>
> The coarse localization module (CLM) is designed to filter out irrelevant objects, thereby enhancing the model efficiency in city-scale visual grounding tasks. However, we must acknowledge that the filter effectiveness can vary depending on the size and attribute of target objects. Large objects (e.g., factories, athletic fields, parking lots) can be easily identified and excluded through the heat map output by CLM. In contrast, small objects (e.g., cars, residential buildings) are more challenging to distinguish, requiring further detailed comparison in the fine-grained matching module (FMM).
>
> We have revised the paper to include a discussion of this phenomenon in **Sec.5 (Limitations)** and plan to address these issues in future work.

---

> ### Author Response · Authors · 2024-11-21
> **Responses to Reviewer P5JV (Part 3)**
>
> **Q6:** Prompt Engineering Requirements: It seems that users must invest effort in engineering their prompts to achieve accurate results, which may pose usability challenges. The authors could consider conducting a user study to gather prompts from a diverse set of users to assess the method’s robustness when different descriptions target the same object.
>
> **A6:** Thanks for your comments. We have performed experiments to analyze how specific parts of text prompts influence grounding performances, which serve as the reference in prompt engineering for different categories of objects. Additionally, we assess the method’s robustness when diverse text prompts from different users target the same object.
>
> **a) Analysis for specific parts of text prompts**
>
> As shown in Figure 6 of the main paper, we visualize the grounding results of systematically removing text descriptions related to color, shape, and neighborhood contextual information. Color description plays an important role in visual grounding and the lack of color information often leads to incorrect results. In contrast, while shape descriptions are less critical for cars, they become more significant for buildings due to the larger variability of shapes. In a scene with many similar objects (e.g., red cars), color and shape descriptions are inadequate to distinguish these objects, and integrating neighborhood contextual information is vital for achieving accurate grounding.
>
> **b) Analysis for diverse text prompts from different users**
>
> As illustrated in Figure 7 of the Appendix, we provide qualitative results for diverse text prompts toward same object. In addition to thorough object feature extraction, the text description plays a crucial role in visual grounding. For the successful grounding cases, text descriptions related color and category are often indispensable.
>
> **Q7:** Progressive Prompting Approach: It would be intuitive to structure prompts in a coarse-to-fine manner—beginning with a broad query that identifies multiple potential candidates and gradually introducing constraints to precisely locate the target.
>
> **A7:** We greatly appreciate the reviewer’s insightful suggestion in the progressive prompting approach, and it holds great potential for enhancing the usability and flexibility in city-scale grounding.
>
> While current CityAnchor employs a two-stage pipeline for coarse localization and fine-grained matching, integrating progressive prompting approach offers a promising area for future work. This process could begin with a broad query to identify multiple potential candidates, followed by progressively refining constraints (such as category, spatial relation, attribute cues) to precisely locate the target object.
>
> For example, consider a city scene where the user queries, “The tall and white-roofed building near the One Stop Building (Landmark)”. In a progressive prompting query, we might have the following steps:
>
> **(1) Initial Prompt (Broad Query):** “The building in this city scene”. The system identifies all the building candidate objects in the city-scale point cloud.
>
> **(2) Refined Prompt (Spatial Constraint):** “The building near the One Stop Building”. The system filters most of the building candidate objects based on their proximity to the “One Stop Building”, progressively localizing the target object.
>
> **(3) Final Prompt (Attribute Matching):** “The tall and white-roofed building”. Additional attributes (e.g., roof color or specific textural features) could further confirm the target object among the filtered building candidate objects.
>
> Progressive prompting aligns naturally with human reasoning and interaction patterns, making it an inspiring way to enhance CityAnchor’s adaptability in real-world scenarios.

---

> ### Comment · Reviewer_P5JV · 2024-11-24
>
> Thank the authors for the detailed and comprehensive response! I have some follow-up questions.
> Re: A2(b) The setting of this experiment is a bit confusing to me. I cannot tell if you are only testing the FMM, or the entire pipeline. My question is, it seems CLM relies on off-the-shelf point cloud seg model a lot, so if CLM cannot identify the novel categories, there's not much FMM can do. A most straightforward way to proof the generalizability of your approach might be something like in Fig.4, but with prompts of novel objects/neighborhood/landmarks.
>
> Also a suggestion for figures like Fig.4, Fig.A1, Fig.A2, it would be better if authors can also draw a target box on CLM results. Right now I'm having a hard time finding the query object on the map (i.e. cannot verify if CLM is able to correctly include the target in the candidates)

---

> > ### Author Response · Authors · 2024-11-25
> > **Responses to Reviewer P5JV**
> >
> > Thanks for valuable comments and suggestions. We evaluate CityAnchor's ability to recognize unknown objects for entire pipeline including CLM and FMM. For better visualization, we have drawn a target box on CLM results for Figure 4, Figure A.1, Figure A.2 and Figure A.9. The PDF has been updated.
> >
> > **CLM has a moderate ability and inherent limitations to identify unknown objects for novel categories.** Image-based open vocabulary segmentation is a challenging task, especially for large-scale maps rasterized from city-scale point clouds. We agree that we may need more efforts to generalize city-scale grounding to open vocabulary level. Though CLM may not always precisely select only the target object into the RoI map in visual grounding for unknown object of novel categories, it can include a broad range of objects potentially matching textual descriptions. This ensures that unknown objects can be considered as the candidate objects for fine-grained matching in FMM.
> >
> > As illustrated in Figure 9 of the Appendix, we add qualitative experiments on RoI detection and visual grounding and provide both the RoI map and grounding results for three representative unknown objects. The final grounding results demonstrate that CityAnchor exhibits the certain generalization capabilities, enabling it to perform 3D visual grounding across a broader range of objects.

---

> ### Author Response · Authors · 2024-12-02
> **Gentle reminder for Reviewer P5JV**
>
> As the author-reviewer discussion period approaches its end, we want to kindly remind you of our responses to your follow-up questions and thoughtful suggestion. We sincerely appreciate your considerate feedback and are eager to address any additional questions or concerns you may have.
>
> If there is any additional information or clarification we could provide to support the discussion, we would be delighted if you could let us know.
>
> Thank you once again for your considerate feedback and the time you have devoted to reviewing our work.

---

> > ### Comment · Reviewer_P5JV · 2024-12-02
> >
> > Thank authors for their thorough explanation and efforts in rebuttal. Majority of my concerns have been addressed, I'll raise my score accordingly.

---

> ### Author Response · Authors · 2024-12-03
> **Response to Reviewer P5JV**
>
> Thank you for the discussion. We sincerely appreciate your time and effort in reviewing our work and helping us improve our paper!

---

### Official Review · Reviewer_YNjM · 2024-11-04

**Soundness:** 2
**Presentation:** 2
**Contribution:** 2
**Rating:** 6
**Confidence:** 3

**Summary:**

The paper addresses the challenge of object localization in city-scale 3D point clouds with a method called CityAnchor, which leverages a multi-modal large language model. Using a two-stage approach of coarse localization and fine-grained matching, CityAnchor efficiently and accurately identifies target objects in large urban scenes. Experimental results show that CityAnchor significantly outperforms existing methods in accuracy and speed, with promising applications in urban planning and geospatial analysis.

**Strengths:**

S1: The method’s coarse-to-fine design is well-structured for efficient large-scale 3D visual grounding.

S2: Experimental results show substantial and objective improvements over existing methods.

S3: The proposed model demonstrates clear practical potential for applications in urban planning and geospatial analysis.

**Weaknesses:**

W1: The method seems heavily reliant on large language models, and the need for manual adjustment of the Region of Interest (RoI) threshold in the coarse localization stage suggests a lack of generalizability.

W2: While the coarse-to-fine localization design is effective, it builds on existing multi-stage approaches without introducing fundamentally new concepts in 3D visual grounding, which suggests that the method's strengths lack a degree of innovation.

Other:
The layout could be improved; for instance, all teaser images should be embedded as PDFs rather than PNGs to ensure the text within images is selectable.

**Questions:**

It’s unclear how the method addresses dynamic objects in urban environments, such as moving vehicles or pedestrians, which could affect the reliability of grounding results.

---

> ### Author Response · Authors · 2024-11-21
> **Responses to Reviewer YNjM**
>
> **We sincerely appreciate the reviewer's insightful comments and time dedicated to evaluating our work. Our responses are listed below.**
>
> **Q1:** The method seems heavily reliant on large language models, and the need for manual adjustment of the Region of Interest (RoI) threshold in the coarse localization stage suggests a lack of generalizability.
>
> **A1:** Thanks for your comments and our responses are in the following.
>
> **a) Reasons for using LLMs.**
>
> LLMs have proven effectiveness across a variety of domains due to their powerful text understanding and reasoning capabilities, and a promising direction to improve visual grounding is to design a multi-modality large language model to process both the text prompts and city-scale 3D point clouds. While the previous city-scale grounding method is trained from scratch using text prompts and 3D point clouds, one of our contributions is to introduce the powerful LLMs in city-scale visual grounding, which outperforms them by 36\% ~ 51\% on Acc@0.25 and 31\% ~ 48\% on Acc@0.50 on two city-scale datasets as shown in Table 1 of the main paper. We believe that our method is aligned with the ongoing trend in the research community to incorporate LLMs into multi-modal applications, where the complementary strengths of language modeling and 3D understanding are combined to tackle complex problems in city-scale scenarios.
>
> **b) We do not manually tune the threshold in CLM.**
>
> In all the experiments conducted on the CityRefer and CityAnchor datasets (e.g., Tables 1, 2, and 3 of the main paper), we employed a fixed RoI threshold (0.3) without adjusting the threshold.
> As shown in Table 1 of the main paper, when the RoI threshold is fixed to 0.3, CityAnchor already performs well and achieve at least 46\% and 35\% on Acc@0.50 on both CityRefer and CityAnchor datasets, respectively.
> Table 4 uses the results of different thresholds to analyze the efficiency and effectiveness of our method but is not our main results.
> We have clarified this in **Sec.4.1.4 (Implementation details)** of the revised paper.
>
> **Q2:** While the coarse-to-fine localization design is effective, it builds on existing multi-stage approaches without introducing fundamentally new concepts in 3D visual grounding, which suggests that the method's strengths lack a degree of innovation.
>
> **A2:** Thanks for the feedback.
> We agree that the coarse-to-fine strategy is widely adopted in various parts of Computer Vision but how to utilize such a strategy in city-scale geospatial analysis is unexplored. Previous method CityRefer randomly selects objects for the visual grounding task, which is slow and inaccurate. Thus, we introduce the coarse-to-fine strategy to improve both accuracy and efficiency.
>
> Designing such a coarse searching module for the city-scale point cloud is also non-trivial. We effectively project the point clouds onto the ground and formulate it as a more manageable 2D coarse-searching problem. All of these techniques are unexplored and a part of our innovation.
>
> **Q3:** The layout could be improved; for instance, all teaser images should be embedded as PDFs rather than PNGs to ensure the text within images is selectable.
>
> **A3:** Thanks for your suggestion. we have adjusted the paper to use PDFs instead of PNGs as the teaser and experiment figures.
>
> **Q4:** It’s unclear how the method addresses dynamic objects in urban environments, such as moving vehicles or pedestrians, which could affect the reliability of grounding results.
>
> **A4:** Thanks for your comments. In our current implementation, we remove dynamic objects, such as moving vehicles, pedestrians, and ongoing constructions, following the baseline method CityRafer. By removing these dynamic objects, we can reduce the risk of grounding errors caused by the unpredictable motion or temporary presence of such objects.
>
> We agree with the reviewer that handling dynamic objects, such as moving vehicles, pedestrians, and active construction sites, is important in the data preprocessing of our method. For future works, it could be an interesting topic to conduct city-scale visual grounding for dynamic 3D point clouds. We have added these discussions in the revision of **Sec.5 (Limitations)**.

---

> ### Author Response · Authors · 2024-11-27
> **Gentle reminder for Reviewer YNjM**
>
> Dear Reviewer YNjM,
>
> As the author-reviewer discussion period is coming to an end, we wanted to kindly remind you of our responses to your valuable comments. We sincerely appreciate your considerate feedback and are eager to address any additional questions or concerns you may have.
>
> If there is any additional information or clarification we could provide to support the discussion, we would be grateful if you could let us know.
>
> Thank you once again for your considerate feedback and the time you have devoted to reviewing our work.

---

> ### Comment · Reviewer_YNjM · 2024-12-01
>
> Thank you for your further explanation. I have raised my scores.

---

> ### Author Response · Authors · 2024-12-03
> **Response to Reviewer YNjM**
>
> Thank you for the discussion. We sincerely appreciate the time and effort you dedicated to reviewing our work and helping us improve the paper!

---

### Official Review · Reviewer_8f4q · 2024-11-04

**Soundness:** 3
**Presentation:** 3
**Contribution:** 3
**Rating:** 6
**Confidence:** 3

**Summary:**

In the paper, the authors introduced CityAnchor, a 3D visual grounding method tailored for city-scale scenes. The CityAnchor is based on a two-stage multi-modal LLM. In the coarse stage, CityAnchor predicts candidate objects that match the query text descriptions on projected 2D maps derived from urban point clouds, allowing us to efficiently filter out redundant regions and concentrate on likely objects. Next, it performs fine-grained matching between text descriptions and the candidate objects to establish the final grounding results by encoding both text descriptions and object features into an LLM. To do that, the authors created a new dataset and evaluated the proposed method on both CityAnchor and CityRefer datasets to demonstrate the effectiveness of the proposed method on city-scale point clouds.

**Strengths:**

First of all, it is interesting that a highly intuitive idea is proposed to address two major issues in existing methods concerning how to improve the multi-modality feature extraction in the large-scale visual grounding and how to efficiently localize the object in a large-scale point cloud.

Although it lacks technical novelty, it effectively implements the necessary modules to address two main problems. More specifically, a coarse location firstly finds possible regions on a projected 2D map of the point cloud while the fine-grained matching accurately determines the most similar region with the given test description.

In experimental section, the proposed method achieved SoTA results in city-level localization tasks, demonstrating its effectiveness. The ablation study is highly analytical for each level module and feature representation.

**Weaknesses:**

One concern is the validity of the coarse localization module. As shown Fig.4, I am curious whether CLM operates as intended. It appears to activate too many candidates, as if it were merely distinguishing between urban and non-urban areas. This raises doubts about whether it significantly aids in the efficient operating in the overall framework.

Experiment analysis and description are not specific and descriptive. For example, a detailed explanation based on specific examples is needed to clarify the novel objects and novel descriptions, whether the reason they are not in the training data is due to an out-of-distribution (OOD) scenario.

**Questions:**

I mentioned all comments including reasons and suggestions in the above sections. I recommend that the author will provide all the concerns, and improve the completeness of the paper. If the rebuttal period resolves the above-mentioned concerns, I will gladly raise my score. Also, there are little vague sentences and grammatical errors in the paper. I recommend that the author will revise the paper.

---

> ### Author Response · Authors · 2024-11-21
> **Responses to Reviewer 8f4q (Part 1)**
>
> **We sincerely appreciate the reviewer's encouraging comments and time dedicated to evaluating our work. We make the responses to each question below.**
>
> **Q1:** One concern is the validity of the coarse localization module. As shown Fig.4, I am curious whether CLM operates as intended. It appears to activate too many candidates, as if it were merely distinguishing between urban and non-urban areas. This raises doubts about whether it significantly aids in the efficient operating in the overall framework.
>
> **A1:** Thanks for your comments. The rasterized map shown in Figure 4 is mainly dominated by small objects, which shows the limited effectiveness of CLM in this case. As illustrated in Figure 2 of the Appendix, we provide additional qualitative results of our method on the CityRefer dataset in "Novel Objects" setting, which shows that CLM can effectively filter our erroneous objects. As illustrated in Figure 6 of the Appendix, we provide heat map visualizations for both simple and complex query texts, which shows that CLM performs reasonably even on complex texts. We have shown in Table 3 of the main paper that CLM significantly reduces computation time while improving grounding accuracy, with 46.86\% of Acc@0.50 and only 32.45s in localizing an object.
>
> The coarse localization module (CLM) is designed to filter out irrelevant objects, thereby enhancing the model efficiency in city-scale visual grounding tasks. However, we must acknowledge that the filter effectiveness can vary depending on the size and attribute of target objects. Large objects (e.g., factories, athletic fields, parking lots) can be easily identified and excluded through the heat map output by CLM. In contrast, small objects (e.g., cars, residential buildings) are more challenging to distinguish, requiring further detailed comparison in the fine-grained matching module (FMM).
>
> We have revised the paper to include a discussion of this phenomenon in **Sec.5 (Limitations)** and plan to address these issues in future work.
>
> **Q2:** Experiment analysis and description are not specific and descriptive. For example, a detailed explanation based on specific examples is needed to clarify the novel objects and novel descriptions, whether the reason they are not in the training data is due to an out-of-distribution (OOD) scenario.
>
> **A2:** We appreciate the reviewer’s insightful feedback and acknowledge the need for more detailed experimental analysis and clearer descriptions regarding novel objects, novel descriptions, and out-of-distribution (OOD) scenarios. In the revised version paper, we have added more specific examples and explanations to address these concerns.
>
> **a) Novel Descriptions (ND).**
>
> "Novel Descriptions" setting indicates that the evaluation objects are seen in the training set, but the query text descriptions are different in test set. This variance in text description may lead to different grounding results, even for the same object.
>
> To clarify "ND" setting, we provide a specific example in **Sec.4.1.3 (Metrics and experimental setting)** of the revised version. For example, a building object may be described in the training set as “A building with white roof near the road”, but in the test set, the same object may be described as “Along with the street, there is a white-roofed building”.
>
> **b) Novel Objects (NO).**
>
> "Novel Objects" setting means the objects in the test set are unseen in the training set. This variance in both object and text description is likely to lead to different grounding results, even for these objects belong the same semantic category (e.g., Building or Car).
>
> To clarify "NO" setting, we provide a specific example in **Sec.4.1.3 (Metrics and experimental setting)** of the revised version. For example, a blue car object may appear in the test set, while the same blue car object and its corresponding text description is not present during training process.
>
> **c) Out-of-distribution (OOD) scenarios.**
>
> To evaluate CityAnchor's ability to recognize "OOD" objects of novel categories, we conducted a generality experiment as illustrated in Figure 8 of the Appendix. In the CityRefer dataset, CityAnchor was trained with only four object categories: Building, Car, Parking, and Ground, and we introduced unknown object categories (such as road Intersection, Woodland, and River) that CityAnchor has not seen before. We evaluated the CityAnchor’s ability to predict the similarity between these unknown objects and both the correct text descriptions (Positive Sample) and incorrect text descriptions (Negative Sample). The grounding results for unknown objects demonstrate that CityAnchor exhibits strong generalization capabilities, enabling it to perform 3D visual grounding across a broader range of objects.

---

> ### Author Response · Authors · 2024-11-21
> **Responses to Reviewer 8f4q (Part 2)**
>
> **Q3:** Also, there are little vague sentences and grammatical errors in the paper. I recommend that the author will revise the paper.
>
> **A3:** Thanks for your comments. We have carefully checked and corrected the vague sentences and grammatical errors in the paper, and the PDF has been updated.

---

> ### Author Response · Authors · 2024-11-27
> **Gentle reminder for Reviewer 8f4q**
>
> Dear Reviewer 8f4q,
>
> As the author-reviewer discussion period is coming to an end, we wanted to kindly remind you of our responses to your valuable comments. We sincerely appreciate your thoughtful feedback and are eager to address any additional questions or concerns you might have.
>
> Please let us know if there's any further information or clarification we can provide to facilitate the discussion process.
>
> Thank you once again for your encouraging comments and time dedicated to evaluating our work.

---

### Official Review · Reviewer_6qAx · 2024-11-06

**Soundness:** 3
**Presentation:** 3
**Contribution:** 3
**Rating:** 8
**Confidence:** 4

**Summary:**

This paper proposes a multi-modal LLM for city-scale 3D visual grounding named CityAnchor. CityAnchor adopts a coarse-to-fine searching strategy. First, a 3D segmentation model generates the potential objects from point clouds. Next, in the coarse localization stage, a LLaVA generates the <RoI> token and a SAM generates the RoI heapmap indicating the candidates of the target object. At last, in the fine-grained matching stage, another LLaVA select the best candidate as the target object. Besides, this work also presents a new dataset for 3D visual grounding. Experiments show that CityAnchor outperforms the previous methods by a large margin.

**Strengths:**

1. The paper is well-written and easy to follow.
2. The experiment results are impressive.
3. The proposed method is well motivated and technically sound.

**Weaknesses:**

1. As also mentioned in Sec.5, the efficiency of the proposed method is a major concern. It involves two LLMs and multiple forward passes in the fine-grained matching stage. Are there any possible solutions to improve the efficiency?
2. How to choose the positive and the negative samples during the training of FMM?
3. The objects are generated by a pretrained 3D segmentation model which can only recognize a closed set of targets. So I expect to see the generality of CityAnchor to unknown objects.
4. Some important details are missing:
- Line 157, what is $T_c$?
- Line 138 & 209, how to determine whether an object has a landmark name?
- Line 206, what is $c$ in $E^s_x$?
- Line 232, which encoder is the concatenated feature vector fed into?

**Questions:**

Please address the questions in the weaknesses section.

---

> ### Author Response · Authors · 2024-11-21
> **Responses to Reviewer 6qAx**
>
> **We sincerely appreciate the reviewer for their invaluable comments and time dedicated to evaluating our work. We provide our responses to each question below.**
>
> **Q1:** As also mentioned in Sec.5, the efficiency of the proposed method is a major concern. It involves two LLMs and multiple forward passes in the fine-grained matching stage. Are there any possible solutions to improve the efficiency?
>
> **A1:** Thanks for your insightful comments. Our method achieves better efficiency than baselines due to the coarse-to-fine strategy. We agree that there is still room for improvement. We provide two possible solutions in the following. We have added this discussion in the revision of **Sec.5 (Limitations)**.
>
> **a) RoI Enhancement.** In the first stage of CityAnchor, the Coarse Localization Module (CLM) is designed to regress the regions of interest (RoI) of target object for filtering out irrelevant objects. We can train the CLM using more diverse datasets to enhance the accuracy of the RoIs, enabling them to comprehend complex query texts, while effectively excluding irrelevant objects and purposefully retaining only the relevant objects.
>
> **b) Landmark Information Guidance.** We observe that landmarks are frequently mentioned in query text, such as “The big oval green ground that is nearby the Perry BarrGreyhound Stadium”. Considering that Retrieval-Augmented Generation (RAG) emerges as a promising solution to mitigate issues of limited expertise in highly specialized queries, we can incorporate landmark information for each object into the CityAnchor as external knowledge using RAG, utilizing landmark information for efficient grounding. Specifically, we plan to select only the objects near the landmark to compare with the query text and avoid too many candidate objects being involved in fine-grained comparisons.
>
> **Q2:** How to choose the positive and the negative samples during the training of FMM?
>
> **A2:** Thanks for your comments. We select positive and negative samples in a ratio of 1:3 for FMM training, and the negative samples are randomly chosen among the candidate objects after excluding the target object. We have included this clarification in **Sec.4.1.4 (Implementation details)**.
>
> **Q3:** The objects are generated by a pretrained 3D segmentation model which can only recognize a closed set of targets. So I expect to see the generality of CityAnchor to unknown objects.
>
> **A3:** Thanks for your comments.
> We follow the experimental setting of CityAnchor to rely on a pre-trained 3D segmentation model to extract objects.
>
> We show that CityAnchor has the potential to generalize to unknown objects. In the revision, we additionally conduct a generality experiment to assess CityAnchor's ability to recognize unknown objects in Figure 8 of the Appendix. On the CityRefer dataset, CityAnchor is trained with only four object categories: Building, Car, Parking, and Ground, and we introduce three objects of unknown object categories (Road Intersection, Woodland, and River) that CityAnchor has not seen before. We show that CityAnchor is still able to determine a reasonable matching score between the text descriptions and the given 3D objects due to the strong generalization ability of LLMs.
>
> **Q4:** Some important details are missing:
>
> Line 157, what is $T_c$?
>
> Line 138 \& 209, how to determine whether an object has a landmark name?
>
> Line 206, what is $c$ in $E^s_x$?
>
> Line 232, which encoder is the concatenated feature vector fed into?
>
> **A4:** Thanks for your comments. We have added these missing details in the revised version.
>
> **a) Line 157, what is $T_c$?**
>
> $T_c$ denotes the output texts (including <RoI> token) in Coarse Localization Module (CLM). We have added a representation of $T_c$ in Figure 2 in revised version paper.
>
> **b) Line 138 \& 209, how to determine whether an object has a landmark name?**
>
> The CityRefer dataset provides landmark names for some objects, with approximately 10\% of the objects already assigned a landmark name, while the remaining objects lack such identifiers. We follow the CityRefer dataset and assign a landmark name to an object based on the geospatial information retrieved from OpenStreetMap.
>
> **c) Line 206, what is $c$ in $E^s_x$?**
>
> $c$ in $E^{s}_x\in \mathbb{R}^{c \times d}$ represents the feature-length of the 2D CLIP feature, which is 256.
>
> **d) Line 232, which encoder is the concatenated feature vector fed into?**
>
> The concatenated feature vector is input into the transformer encoder of the Fine-grained Matching Module (FMM), which is fine-tuned from LLaVA. The FMM predicts the similarity between the query text and the objects based on the concatenated feature vector and then selects the object with the largest similarity as the final grounding result for the query text.

---

> ### Author Response · Authors · 2024-11-27
> **Gentle reminder for Reviewer 6qAx**
>
> Dear Reviewer 6qAx,
>
> As the author-reviewer discussion period is coming to a close, we would like to provide a gentle reminder that we have posted a response to your valuable comments. We sincerely appreciate your encouraging feedback and are eager to address any additional questions or concerns you may have.
>
> If there is any additional information or clarification we might provide to support the discussion, we would be grateful if you could let us know.
>
> Thank you once again for your encouraging comments and time dedicated to evaluating our work.

---

> > ### Comment · Reviewer_6qAx · 2024-12-03
> >
> > Thanks the authors for the detailed explanations and most of my concerns have been addressed. In spite of the potential efficiency problem caused by the usage of LLMs, I still think this work has made enough contribution to the community. For this reason, I would keep my rating unchanged.

---

> ### Author Response · Authors · 2024-12-03
> **Response to Reviewer 6qAx**
>
> Thank you for the discussion. We really appreciate your efforts in reviewing our work and the insightful comments that help us improve the paper!

---

### Author Response · Authors · 2024-11-21
**Overall Response from Authors**

We sincerely appreciate all reviewers for their detailed feedback and constructive suggestions. We are glad that the reviewers recognize the effectiveness of CityAnchor in addressing the 3D city-scale visual grounding problem with positive comments like the well-structured coarse-to-fine design (6qAx, 8f4q, YNjM, P5JV), the impressive experimental results demonstrating significant performance improvements (6qAx, 8f4q, YNjM, P5JV) compared to the existing baseline methods, and the introduction of the CityAnchor dataset can serve as a valuable resource for future research (P5JV).

After carefully improving the quality of our submission, we present here a revised main paper and supplementary materials, and the pdf has been updated. The modifications in the main paper have been highlighted in red, the added experiments and necessary analysis supplemented in the appendix have been highlighted in blue.

Changes in the main paper include:

$\bullet$ Detailed meaning of $c$ and encoding process for FMM in Sec.3.3.1.

$\bullet$ Improved writing: StreetSgin $\rightarrow$ StreetSign in Sec.4.1.1.

$\bullet$ Specific examples added for "Novel Objects" and "Novel Descriptions" settings in Sec.4.1.3.

$\bullet$ Clarification of fixed RoI threshold (0.3) and positive and negative sample ratio (1:3) in Sec.4.1.4.

$\bullet$ More in-depth discussion for limitations and future work in Sec.5.

Changes in the revised supplementary material include:

$\bullet$ Additional Results (Figure A.2) and discussion on CityRefer dataset with RoI map in Sec.A.1.

$\bullet$ Results (Figure A.7) and analysis on diverse text prompts in Sec.A.4.

$\bullet$ Generality experiment (Figure A.8 and Figure A.9) and discussion for unknown objects in Sec.A.5.

---

### Meta-Review · Area_Chair_sQoy · 2024-12-22

**Metareview:**

This paper introduces an approach that localizes an object in a city-scale point cloud. The key of the proposed idea has two stages: one that roughly locates possible positions in the 2D map of the point cloud using LLM, and the next step accurately localizes these potential candidates. The authors conduct experiments with the CityRefer dataset and the newly annotated synthetic dataset, and the authors demonstrate superior accuracy on these two datasets. Given the consensus from reviewers for the paper's acceptance, AC confirms that the proposed approach introduced interesting ideas in this field and proposed a challenging dataset for the task.

**Additional Comments On Reviewer Discussion:**

Overall, all reviewers were inclined toward paper acceptance after the rebuttal phase. AC confirms that the discussion was constructive, and the authors provided solid feedback. Finally, the authors updated the main paper and supplementary materials correctly.

More specifically, reviewer 6qAx asks a question regarding the efficiency of the proposed method, and the authors mention possible ways to improve the speed by using RoI enhancement and landmark information guidance. The authors also provided answers regarding the question about positive and negative sampling ratios, and the authors explained CityAnchor settings that use known object classes. The other comments are mostly about equations and network design. The reviewer 8f4q mentioned the validity of the coarse localization module, such as CLM, in the proposed approach, and the authors provide additional results and note the limitation of the proposed approach. The reviewer 8f4q requested a more informative discussion regarding the analysis of the experimental results. Although reviewer 8f4q did not confirm the author's feedback, AC confirms that the authors correctly addressed the concerns. The reviewer YNjM provides a short review regarding the motivation for using LLMs, threshold setting in CLM, inserted image format, and moving objects. The authors provided feedback on the comments, and the reviewer YNjM raised the score in the end. The reviewer P5JV provided a thorough review. The questions were about the limitation of the 2D mapping, dependence of the point segmentation model, and justification of the proposed dataset, inference time, RoI threshold, prompt engineering requirements. The reviewer P5JV also suggests the constructive direction that adjusts the structure of the prompts in a coarse-to-fine manner. There was another discussion about identifying unknown objects. The reviewer P5JV finally raised the score, clarifying that all major concerns have been resolved.

---

### Decision · Program_Chairs · 2025-01-22

Accept (Poster)